# DMWM: Dual-Mind World Model with Long-Term Imagination

**Lingyi Wang**[1]* **Rashed Shelim**[1] **Walid Saad**[1] **Naren Ramakrishnan**[2]

[1] Department of Electrical and Computer Engineering, Virginia Tech, USA
[2] Department of Computer Science, Virginia Tech, USA
Emails: {lingyiwang, rasheds, walids, naren}@vt.edu

## Abstract

Imagination in world models is crucial for enabling agents to learn long-horizon policies in a sample-efficient manner. Existing recurrent state-space model (RSSM)-based world models depend on single-step statistical inference to capture the environment dynamics, and, hence, they cannot effectively perform long-term imagination tasks due to the accumulation of prediction errors. Inspired by the dual-process theory of human cognition, we propose a novel dual-mind world model (DMWM) framework that integrates logical reasoning to enable imagination with logical consistency. DMWM is composed of two components: an RSSM-based System 1 (RSSM-S1) component that handles state transitions in an intuitive manner and a logic-integrated neural network-based System 2 (LINN-S2) component that guides the imagination process through hierarchical deep logical reasoning. The inter-system feedback mechanism is designed to ensure that the imagination process follows the logical rules of the real environment. The proposed framework is evaluated on benchmark tasks that require long-term planning from the DMControl suite and the robotic platforms. Extensive experimental results demonstrate that the proposed framework yields significant improvements in terms of logical coherence, trial efficiency, data efficiency and long-term imagination over the state-of-the-art world models. The code is available at https://github.com/news-vt/DMWM.

## 1 Introduction

Imagination is a core capability of world models that allows agents to predict and plan effectively within internal virtual environments by using real-world knowledge [1, 2, 3, 4]. By predicting future scenarios in latent spaces, agents can evaluate potential outcomes without frequent real-world interactions thereby significantly improving data efficiency and minimizing trial-and-error costs. For complex tasks requiring long-term planning, imagination capabilities allow world models to evaluate the long-term consequences of diverse strategies and identify the optimal action plans. Consequently, the effectiveness of model-based decision-making approaches, such as model-based reinforcement learning (RL) [5, 6, 7, 8] and model predictive control (MPC) [9, 10, 11], can heavily depend on the quality of their imagination abilities.

One of the most widely used frameworks for world models is the so-called recurrent state-space model (RSSM) [2, 9, 12] and its variants [13, 14, 15, 16] that combine deterministic recurrent structures with stochastic latent variables to model environmental dynamics in a compact latent space. By doing so, RSSM models can capture the sequential dependencies and uncertainty of their target environment. However, RSSM cannot provide reliable, long-term predictions over extended imagination horizons due to the accumulation of prediction errors and the limitations of statistical inference [17, 18, 19]. In particular, although existing RSSM-based solutions can generate accurate short-term predictions

---

*Corresponding author.

39th Conference on Neural Information Processing Systems (NeurIPS 2025).

in a single-step manner, small errors inevitably propagate over longer time horizons [20], gradually amplifying over each step and resulting in significant deviations between the imagined and actual states. Moreover, RSSM schemes often optimize state-space representations through reconstruction loss or regression. This approach can lead to overfitting to observed patterns and cannot properly capture latent dynamics [18, 20, 21], particularly in complex, dynamic environments.

Several models [17, 18, 20, 22, 23, 24] have been proposed for long-term planning. For instance, trajectory-based models [20, 23] learn long-term dynamics by directly predicting future states to reduce compounding errors compared to single-step prediction models. Latent space-enhanced models [17, 18] incorporate abstracted long-term information, such as high-level goals and global knowledge, to mitigate error propagation in each single-step prediction. However, all of these approaches [17, 18, 20, 22, 23] still rely on statistical inference, which inevitably leads to prediction error accumulation and drift over long horizons. By skipping intermediate reasoning steps, these approaches [17, 18, 24] lose the logical consistency and interpretability required for complex, high-precision planning. Hence, existing approaches cannot provide robust and reliable imagination over an extended horizon size.

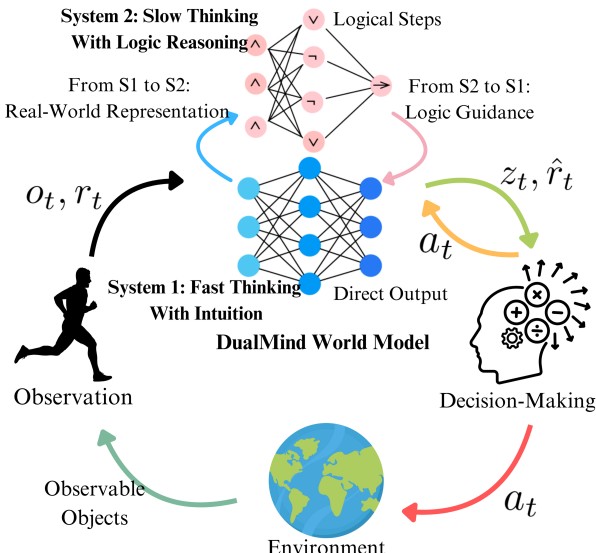

Figure 1: The proposed framework for DMWM.

Motivated by the challenges of long-term imagination and the limitations of existing long-term planning approaches, the main contribution of this paper is a novel dual-mind world model (DMWM) framework based on the dual-process theory of human cognition [25, 26, 27]. The proposed DMWM framework can achieve reliable and efficient imagination by synergistically combining the complementary strengths of System 1 and System 2 learning processes [28, 29]. By designing this framework shown in Figure 1, we make the following key contributions:

- We propose DMWM, a novel world model framework that integrates the dual-process theory of human cognition (System 1 and System 2) to endow agents with robust, long-term imagination capabilities driven by both data and logical reasoning. Particularly, based on the RSSM-based System 1 (RSSM-S1) component that predicts the state transitions in a fast, intuitive-driven manner, we further propose logic-integrated neural network-based System 2 (LINN-S2) component that is used, for the first time, to guide imagination through logical reasoning at a higher level.

- In LINN-S2, we introduce logical regularization rules to conduct logical reasoning within the state and action spaces by using operations such as $\wedge$, $\vee$, $\neg$ and $\rightarrow$. The logical rules allows the logical consistency and interpretability of the world model. Additionally, we propose a new recursive logic reasoning framework that extends local reasoning into globally consistent long-term planning, enabling the modeling of logical sequence dependencies in complex tasks.

- We design an effective inter-system feedback mechanism. In particular, LINN-S2 provides logical constraints to guide RSSM-S1 so as to ensure that predicted sequences are consistent with domain-specific logical rules. For the feedback based on real-world observations and latent representations from RSSM-S1, it updates the domain-specific logic of LINN-S2, thereby allowing dynamic refinement and adaptation. This novel inter-system feedback mechanism can thus allow the human-like, dual-process cognitive abilities for agents.

- We evaluate the proposed DMWM with actor-critic based RL and MPC in extensive experiments including DMControl and robotic tasks. Simulation results demonstrate that DMWM is able to respectively provide 14.3%, 5.5-fold, 32% and 120% improvement in logic consistency, trial efficiency, data efficiency and reliable imagination over an extended horizon size compared to baselines in complex tasks.

## 2 Proposed DMWM Framework

In this section, we introduce the proposed DMWM framework inspired by the dual-process theory of human cognition. DMWM consists of RSSM-S1 and LINN-S2. First, we introduce RSSM-S1, which builds upon the RSSM architecture to learn the environment dynamics in a latent space and perform fast, intuitive state representations and predictions. Next, we introduce a novel LINN-S2 to capture the intricate logical relationships between the state space and the action space. By employing a hierarchical deep reasoning framework, LINN-S2 facilitates structured reasoning and enforces logical consistency over extended horizons. Finally, we explain the proposed inter-system feedback mechanism. The pipeline of the proposed DMWM framework is shown in Figure 1.

### 2.1 RSSM-based System 1

The RSSM-S1 component is based on DreamerV3 [7], which is represented by

$$
\begin{aligned}
\text{Deterministic State:} \quad & h_t = f_\varphi\left(h_{t-1}, z_{t-1}, a_{t-1}\right), \\
\text{Encoder:} \quad & z_t \sim q_\varphi\left(z_t \mid h_t, o_t\right), \\
\text{Stochastic State:} \quad & \hat{z}_t \sim p_\varphi\left(\hat{z}_t \mid h_t\right), \\
\text{Reward Predictor:} \quad & \hat{r}_t \sim p_\varphi\left(\hat{r}_t \mid h_t, z_t\right), \\
\text{Decoder:} \quad & \hat{o}_t \sim p_\varphi\left(\hat{o}_t \mid h_t, z_t\right),
\end{aligned}
\tag{1}
$$

with the deterministic state $h_t$, observation $o_t$, predicted observation $\hat{o}_t$, stochastic state $z_t$, predicted stochastic state $\hat{z}_t$, action $a_t$ and the predicted reward $\hat{r}_t$ at time step $t$. RSSM-S1 achieves an effective balance between deterministic and stochastic states and enables efficient data-driven prediction similar to the intuitive and automatic processes of System 1. The deterministic state captures data patterns, and the stochastic state models inherent uncertainty and dynamics for complex environments.

**System 1 loss.** RSSM-S1 is optimized by using the loss function of DreamerV3 [7]:

$$
\begin{aligned}
\mathcal{L}_{\text{S1}}(\varphi) &= \mathcal{L}_{\text{pred}}(\varphi) + \varpi_{\text{dyn}} \mathcal{L}_{\text{dyn}}(\varphi) + \varpi_{\text{rep}} \mathcal{L}_{\text{rep}}(\varphi), \\
\mathcal{L}_{\text{pred}}(\varphi) &= -\ln p_\varphi\left(o_t \mid z_t, h_t\right) - \ln p_\varphi\left(r_t \mid z_t, h_t\right), \\
\mathcal{L}_{\text{dyn}}(\varphi) &= \text{KL}\left[\text{sg}\left(q_\varphi\left(z_t \mid h_t, o_t\right)\right) \| p_\varphi\left(z_t \mid h_t\right)\right], \\
\mathcal{L}_{\text{rep}}(\varphi) &= \text{KL}\left[q_\varphi\left(z_t \mid h_t, o_t\right) \| \text{sg}\left(p_\varphi\left(z_t \mid h_t\right)\right)\right],
\end{aligned}
\tag{2}
$$

where $\varpi_{\text{dyn}}$ and $\varpi_{\text{rep}}$ are respectively the weight factors of the dynamic loss $\mathcal{L}_{\text{dyn}}$ and the representation loss $\mathcal{L}_{\text{rep}}$, and $\text{sg}(\cdot)$ represents the stop-gradient operator.

**System 1 limitations.** The RSSM framework can be viewed as a computational counterpart of the System 1 component in the dual-process theory of cognition [25] characterized by fast, intuitive-driven reasoning. However, it is inherently constrained by two critical limitations: *the absence of explicit logical reasoning* and *the inability to maintain coherence over extended temporal horizons*. While RSSM-S1 enables efficient and immediate response through pattern recognition and intuitive prediction, these strengths come at the cost of enforcing logical consistency or inferring causal relationships in complex scenarios. Moreover, RSSM-S1's focus on short-term processing limits its ability to

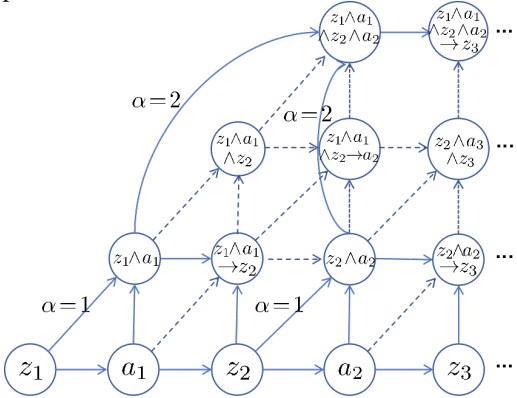

Figure 2: Logic reasoning for LINN-S2.

integrate information across long periods, resulting in fragmented or inconsistent predictions in tasks that demand global planning or long-term foresight. These inherent limitations demonstrate the need to complement the capabilities of RSSM-S1 with mechanisms of structured reasoning and sustained temporal coherence, similar to System 2, to construct a more robust, cognitive world model framework.

### 2.2 Logic-integrated neural network-based System 2

Next, we introduce the LINN-S2 component whose deep logical reasoning is shown in Figure 2. In LINN-S2, the states and actions are encoded as logic vector inputs, enabling logical deduction through operations such as negation ($\neg$), conjunction ($\wedge$), disjunction ($\vee$), and implication ($\rightarrow$). To establish these logical operations, the LINN framework [30] serves as the foundational module for

System 2. We propose a novel hierarchical logical reasoning framework and extended regularization rules for implication operations to capture and reason about structural relationships between the state and action spaces, thus endowing the world model with logical inference capability.

**Neural modules for basic logic operations.** In logical reasoning for world models, it is essential to uncover structural information and semantic relationships across different embedding spaces, specifically the action space and the state space. In [30], the authors proposed a straightforward concatenation-based method for LINN. However, the approach of [30] is limited by the fact that it limits that input vectors are from the same source space. Moreover, the straightforward concatenation-based method overlooks the semantic disparities between states and actions, risking the loss of critical information and failing to capture complex cross-space logical relationships.

To address the need for cross-space logical reasoning in world models, we propose to explore the action embeddings and apply the Kronecker product for cross-space feature alignment, which preserves logic integrity and captures second-order relationships. The (imagined) state logical embedding $v$ and the state logical embedding $m$ are obtained with multilayer perception (MLP) by

$$v = \boldsymbol{W}_2^s f\left(\boldsymbol{W}_1^s(z) + \boldsymbol{b}_w^s\right), \quad m = \boldsymbol{W}_2^a f\left(\boldsymbol{W}_1^a(z \oplus a) + \boldsymbol{b}_w^a\right), \tag{3}$$

where $\oplus$ is the operation of vector concatenation, and $z \oplus a$ aims to capture the action logic within the state context. The formulations of operations $(\text{AND}, \text{OR}, \text{NOT})$ are respectively given by

$$\text{AND}(v, m) = \boldsymbol{W}_2^a f\left(\boldsymbol{W}_1^d(v \oplus m) + \boldsymbol{b}_w^d\right) + \text{Conv2D}(v \otimes m, \boldsymbol{K}^d) + \boldsymbol{b}_k^d, \tag{4}$$

$$\text{OR}(v, m) = \boldsymbol{W}_2^o f\left(\boldsymbol{W}_1^o(v \oplus m) + \boldsymbol{b}_w^o\right) + \text{Conv2D}(v \otimes m, \boldsymbol{K}^o) + \boldsymbol{b}_k^o, \tag{5}$$

$$\text{NOT}(v) = v + \boldsymbol{W}_2^n f\left(\boldsymbol{W}_1^n v + \boldsymbol{b}_w^n\right), \tag{6}$$

where $v \in \mathbb{R}^d$, $m \in \mathbb{R}^d$, $\boldsymbol{W}_1^l \in \mathbb{R}^{d \times 2d}$, $\boldsymbol{W}_2^l \in \mathbb{R}^{d \times d}$, $\boldsymbol{b}_w^l \in \mathbb{R}^d$, $\boldsymbol{b}_k^l \in \mathbb{R}^d$ are the parameters of the logical neural networks, $l \in \{d, o, n\}$, $\text{Conv2D}(\cdot)$ represents the convolution neural network, $\boldsymbol{K}$ is the convolution core, $\otimes$ is the operation of Kronecker product, and $f(\cdot)$ is the activation function.

Logical operations capture intricate logical correlations that cannot be fully encapsulated by the geometric properties of the embedding space. For instance, the logical independence in $\text{NOT}(v)$, reflected in the relationship between $v$ and $\neg v$, does not correspond precisely to vector orthogonality. Logical operations are governed by logical axioms (e.g., identity, annihilator, and Complement), which impose algebraic constraints that transcend purely geometric interpretations, which are introduced in the logic regularization part.

**Implication relationship for state reasoning.** Based on the basic logical operations $(\text{AND}, \text{OR}, \text{NOT})$, we implement the implication operation $(p_\varphi \to q_\varphi)$ to enable logical reasoning within the state and action logical embedding spaces. The implication operation is critical for assessing the rationality of a predicted state given the current imagination and action embeddings, which is derived through the equivalence relationship $p \to q \iff \neg p \lor q$. Hence, the operation IMPLY for $v \to m$ is realized based on the fundamental operations of negation $(\neg)$ and conjunction $(\wedge)$, represented by

$$\text{IMPLY}(v, m) = \text{OR}(\text{NOT}(v), m). \tag{7}$$

Table 1: Partital Logical Regularizers and Rules for Implication $\to$

| Logical Rule | Equation | Logic Regularizer $r_i$ |
|---|---|---|
| **Identity** | $w \to \mathbf{T} = \mathbf{T}$ | $r_{11} = \sum_{w \in W} 1 - \text{Sim}(\text{OR}(\text{NOT}(w), \mathbf{T}), \mathbf{T})$ |
| **Annihilator** | $w \to \mathbf{F} = \neg w$ | $r_{12} = \sum_{w \in W} 1 - \text{Sim}(\text{OR}(\text{NOT}(w), \mathbf{F}), \text{NOT}(w))$ |
| **Idempotence** | $w \to w = \mathbf{T}$ | $r_{13} = \sum_{w \in W} 1 - \text{Sim}(\text{OR}(\text{NOT}(w), w), \mathbf{T})$ |
| **Complement** | $w \to \neg w \equiv \neg w$ | $r_{14} = \sum_{w \in W} 1 - \text{Sim}(\text{OR}(\text{NOT}(w), \text{NOT}(w)), \text{NOT}(w))$ |

**Logical regularizations.** To ensure that neural modules accurately perform logical operations, the work in [31] introduced logic regularizers $\{r_i\}$ that enforce adherence to fundamental logical rules, thereby constraining module behavior. While neural networks can implicitly learn logical operations from data, the explicit incorporation of logical constraints improves model consistency, interpretability, and robustness while maintaining efficient neural computation. Derived from principles of $\text{AND}, \text{OR}$, and $\text{NOT}$, logic regularizers establish a unified inference framework with fundamental and advanced operations. This approach aligns model outputs with human-understandable reasoning, bridging neural networks and formal logic, and enabling effective learning, representation, and inference of logical formulas for interpretable reasoning in world models.

We extend the foundational framework of logical regularization in [30] by introducing implication regularizers to address advanced logical rules. As summarized in Table 1, the implication rules refine basic logical structures to handle intricate constructs. By explicitly encoding principles such as *contraposition* ($w \to v \equiv \neg v \to \neg w$) and *complement* ($w \to \neg w \equiv \neg w$), LINN-S2 improves its capacity to manage compound logical expressions while ensuring consistency across implication operations. The extended logical regularizers for IMPLY ($r_{11}$-$r_{14}$) are given in Table 1, combined with the basic logical regularizers for AND, OR, and NOT ($r_1$-$r_{10}$), collectively define the regularization loss function $\mathcal{L}_{\text{reg}} = \frac{1}{N_r} \sum_i r_i$, where $r_i$ represents individual logical regularizers and $N_r$ presents the total number of the regularizers. The complete table of the logical regularizers is given in Appendix G.

**Proposed symbolic representation of hierarchical logical reasoning.** We now present the symbolic representation of LINN-S2 for deep logical reasoning. Specifically, the logical reasoning process is formalized by using symbolic logic to represent logical relationships in sequences. By leveraging logical operations ($\wedge, \to$), LINN-S2 constructs a hierarchical logical reasoning framework that facilitates systematic and interpretable reasoning over sequential dependencies. The framework of the hierarchical logical reasoning is illustrated in Figure 2, which includes three key procedures given as follows.

*1) Local logical composition*: Each (imagined) state logical embedding $v_t$ and action logical embedding $m_t$ are combined by using the logical conjunction ($\wedge$) to capture the intrinsic logic as

$$c_t : v_t \wedge m_t, \quad \forall t < T. \tag{8}$$

The composition $c_t$ establishes localized logical features based on the existing $v_t$ and $m_t$.

*2) Recursive implication reasoning*: To ensure logical consistency across (imagined) states, each composition ($c_t$) undergoes an implication operation ($\to$) that aligns the logical information with the subsequent (imagined) state

$$\phi_t : c_t \to z_{t+1}, \quad \forall t < T. \tag{9}$$

The recursive formulation (9) encodes sequential dependencies and enforces consistency in predictive state transitions across the hierarchy. However, $\phi_t$ provides only single-step logical inference, and it lacks reasoning depth and comprehensiveness. To address this limitation, we propose incorporating deterministic historical information into the logical reasoning framework. The deep recursive implication reasoning is represented by

$$\phi_t^\alpha : c_{t-\alpha} \cdots \wedge c_{t-1} \wedge c_t \to z_{t+1}, \forall \alpha < t < T, \tag{10}$$

where $\alpha$ is the inference depth. In complex scenarios with long-term dependencies, the model utilizes logical relationships from past states to ensure coherent reasoning. Capturing logic across time steps retains historical information, strengthens sequential dependencies and enhances global consistency. This prevents inference bias from information loss during multi-step reasoning and ensures the reliability of long-term imagination.

*3) Global logical chain*: The global reasoning process integrates local consistency and recursive reasoning into a unified global logical chain $L_T$ within the time period $T$

$$L_T^\alpha = \phi_1^\alpha \wedge \phi_2^\alpha \wedge \phi_3^\alpha \cdots \phi_{T-1}^\alpha \to \mathbf{T}, \tag{11}$$

where $\mathbf{T}$ represents the consistency condition, ensuring that the (imagined) state sequences align with the reasoning objectives. This formulation consolidates local information into a globally consistent reasoning framework.

In this way, LINN-S2 enhances the formal interpretation and mathematical rigor of world models along with several key features. First, the hierarchical structure encodes deep logical reasoning as a layered process, which allows logical consistency between state and action representations. Second, logical composition leverages logical operations for binding states and actions in a logical manner. This composition facilitates formal logic principles and enables modular, interpretable reasoning. Finally, the recursive implication ensures robustness and interpretability for long-term imagination.

**System 2 loss.** The loss function of the logic reasoning with inference depth $\alpha$ is given by

$$\mathcal{L}_{\text{log}}^\alpha = \frac{1}{T-1} \sum_t \text{Sim}(\phi_t^\alpha, \mathbf{T}) - \text{Sim}(\phi_t^\alpha, \mathbf{F}) = \frac{1}{T-1} \sum_t \text{Sim}(\phi_t^\alpha, \mathbf{T}) - \text{Sim}(\phi_t^\alpha, \text{NOT}(\mathbf{T})), \tag{12}$$

where the function Sim is the logic similarity metric that takes value between 0 and 1. In practice, $\mathbf{T}$ is a randomized fixed vertor, and $\mathbf{F}$ is obtained by $\mathrm{NOT}(\mathbf{T})$. We use the cosine similarity since the logical information of action and state spaces has been aligned in the logical vector space, given by

$$\mathrm{Sim}(v, m) = \sigma\left(\kappa(v \cdot m)/(\|v\|\|m\|)\right). \tag{13}$$

To ensure the logical consistency of basic operations of AND and OR by order-independence, i.e., $v \wedge m = m \wedge v$ and $v \vee m = m \vee v$, the inputs of AND and OR are randomly disrupted. Moreover, the $\ell_2$-regularization term $\mathcal{L}_v$ [30] is employed to prevent the vector lengths from exploding, which could otherwise lead to trivial solutions (e.g., logical rules becoming ineffective) during optimization, and constraint on the model parameters to mitigate the risk of overfitting, represented by

$$\mathcal{L}_{\ell_2} = \sum_{v \in \mathcal{V}} \|v\|_F^2 + \sum_{m \in \mathcal{M}} \|m\|_F^2 + \sum_{w \in \mathcal{W}} \|w\|_F^2, \tag{14}$$

where $\mathcal{W}$ is the model parameter of LINN-S2. The loss function of System 2 is expressed as

$$\mathcal{L}_{\mathrm{S2}}(w) = \sum_{\alpha=0}^{\Lambda} \mathcal{L}_{\log}^{\alpha} + \beta_{\mathrm{reg}} \mathcal{L}_{\mathrm{reg}} + \beta_{\ell_2} \mathcal{L}_{\ell_2}, \tag{15}$$

where $\beta_{\mathrm{reg}}$ and $\beta_{\ell_2}$ represents the weight factors, and $\Lambda$ denotes the maximum reasoning depth.

## 2.3 Inter-System Feedback Mechanism

Next, we develop a novel inter-system feedback mechanism to enable communications of observation signals and logical signals between System 1 and System 2.

**Feedback from S1 to S2.** During the real-environment interactions, LINN-S2 updates the domain-specific logical relationships by using the actual state transitions from RSSM-S1. Particularly, the state-action sequence $\{z_t, a_t, z_{t+1}\}_{t=0}^{T}$ captured by RSSM-S1 based on observations $\{o_t\}_{t=0}^{T}$ serves as labeled data, which is fed into LINN-S2 with the objective of minimizing the inference loss (12).

**Feedback from S2 to S1.** To embed LINN-S2-inspired logical reasoning into RSSM-S1, we propose a logic feedback that utilizes LINN-S2's logical consistency mechanisms to guide RSSM-S1. By unifying high-level logical structures with low-level representations, the proposed approach fosters a more robust and coherent world model for imagination. We particularly introduce the logical rules and rederive the variational evidence lower bound (ELBO) [5] with a logic inference term by

$$p_{\varphi}(o_{1:T}, z_{1:T} \mid a_{1:T}) = \prod_{t=1}^{T} p_{\varphi}(o_t \mid z_t)\tilde{p}_{\phi}(z_t \mid z_{t-1}, a_{t-1}), \tag{16}$$

where $\tilde{p}_{\phi}(z_t \mid z_{t-1}, a_{t-1}) \propto p_{\varphi}(z_t \mid z_{t-1}, a_{t-1}) \cdot \mathcal{C}(\phi(z_t, z_{t-1}, a_{t-1}))$ and $\mathcal{C}(\phi) = \mathrm{Sim}(\phi, \mathbf{T})$ is the logical consistency function that measures whether the logical rule $\phi$ is satisfied by imagination. The logical ELBO of the observation loss can be expressed as (Derivation in Appendix C)

$$\ln p_{\varphi}(o_{1:T} \mid a_{1:T}) = \int p_{\varphi}(o_{1:T}, z_{1:T} \mid a_{1:T}) \, dz_{1:T} \geq \sum_{t=1}^{T} (\underbrace{\mathbb{E}_{q_1}\left[\ln p_{\varphi}(o_t \mid z_t)\right]}_{\text{Decoding}}$$
$$+ \underbrace{\mathbb{E}_{q_1}\left[\ln \mathcal{C}(\phi(z_t, z_{t-1}, a_{t-1}))\right]}_{\text{Logic Inference}} - \underbrace{\mathbb{E}_{q_2}\left[\mathrm{KL}\left[q_{\varphi}(z_t \mid o_{\leq t}, a_{<t}) \| p_{\varphi}(z_t \mid z_{t-1}, a_{t-1})\right]\right]}_{\text{Prediction}}). \tag{17}$$

where $q_1 = q_{\varphi}(z_t \mid o_{\leq t}, a_{<t})$, $q_2 = q_{\varphi}(z_{t-1} \mid o_{\leq t-1}, a_{<t-1})$, and the state priors can be approximately obtained by past observations and actions $q_{\varphi}(z_{1:T} \mid o_{1:T}, a_{1:T}) = \prod_{t=1}^{T} q_{\varphi}(z_t \mid h_t, o_t)$.

With the proposed inter-system feedback, DMWM enables the agents to think in a human-like, dual-process cognitive way for more robust, reliable and long-term imagination.

## 3 Experimental Results and Analysis

In this section, we conduct extensive experiments to address the following key questions: (a) Can our model effectively capture logical relationships in dynamic environments?, (b) Does enhanced logical consistency enable our model to achieve higher task rewards under limited environment trials and data?, and (c) Over an extended horizon, can our model generate reliable long-term imagination?

**Experimental setup.** The training environments consist of 20 continuous control tasks from DeepMind control (DMC) suite, 4 robotic tasks from ManiSkill2 platform, and 4 robotic tasks from

MyoSuite platform. We evaluate DMWM using two model-based decision-making approaches: An actor-critic reinforcement learning method and a gradient-based model predictive control (Grad-MPC) approach [11], referred to as DMWM-AC and DMWM-GD, respectively. Training details for DMWM-AC and DMWM-GD are provided in Appendix D. For comparison purposes and benchmarking, we include DreamerV3 [7], Dreamer-enabled Grad-MPC [11], and TD-MPC2 [13] as baselines. To highlight the limitations of using a single RSSM for world cognition, we compare our method against two state-of-the-art RSSM variants: Hieros [3] and HRSSM [15]. HRSSM improves representation robustness via masking and bisimulation to mitigate visual noise interference, while Hieros enhances long-term modeling and exploration efficiency through S5WM and hierarchical strategies. Details on environment and model settings are provided in Appendix E.

**Logical consistency.** Figure 3 shows the logical correlations between individual state-action pairs $(s_i \wedge a_j)$ and the target state $(s_{30})$. The diagonal shows strong correlations for one-step pairs $s_i \wedge a_i \rightarrow s_{30}$, which is the key path of localized reasoning. The off-diagonal elements show interdependency across different state-action pairs $(s_i \wedge a_j, i \neq j)$ that propagate global logical information. The observed patterns highlight the need for deep logical reasoning to capture both short-term and long-term logical dependencies.

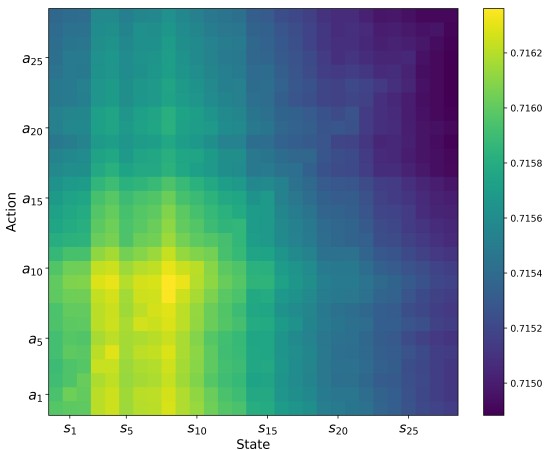

Figure 3: Heatmap of deep logic correlations for sequential imagination $s_0 \wedge a_0 \wedge ...s_{29} \wedge a_{29} \rightarrow s_{30}$ with reasoning depth $\alpha = 30$. The horizontal axis indicates the past states $s_i$ and the vertical axis indicates past actions $a_j$. The color of points represents the logical strength of long-term state-action pairs.

Table 2 compares the logical consistency of the proposed DMWM against various baselines on DMC tasks. Logical consistency data for 20 tasks with different horizon sizes is provided in Appendix H.1. The proposed DMWM achieves state-of-the-art logical consistency in both mean logic loss and stability across all DMC environments. DMWM respectively achieves 14.3%, 2.6% and 3.3% improvement in logic consistency compared to Dreamer, Hieros, and HRSSM. For instance, while the masking and hierarchical strategies in Hieros and HRSSM can reduce single-step propagation errors by mitigating environment noise, they still have difficulty in addressing long-term imagination due to predictive deviation in statistical inference and error accumulation. This highlights the limitations of relying solely on System 1 and the need for logical consistency from System 2 for robust imagination.

Table 2: Performance comparison of our approach with various baselines on DMC tasks in terms of logic consistency. The mean and variance of logical consistency are reported over 100 test episodes with the horizon size $H = 30$. Complete results of logical consistency over 20 tasks with varying horizon size are concluded in Appendix H.1.

| Env | Dreamer [7] | Hieros [3] | HRSSM [15] | DMWM (**Proposed**) |
|---|---|---|---|---|
| Cartpole Balance | $0.683 \pm 0.057$ | $0.711 \pm 0.032$ | $0.713 \pm 0.041$ | $\mathbf{0.727 \pm 0.023}$ |
| Pendulum Swingup | $0.611 \pm 0.137$ | $0.709 \pm 0.054$ | $0.699 \pm 0.079$ | $\mathbf{0.730 \pm 0.037}$ |
| Reacher Hard | $0.608 \pm 0.121$ | $0.702 \pm 0.062$ | $0.703 \pm 0.072$ | $\mathbf{0.730 \pm 0.042}$ |
| Finger Turn Hard | $0.627 \pm 0.131$ | $0.698 \pm 0.061$ | $0.703 \pm 0.073$ | $\mathbf{0.725 \pm 0.029}$ |
| Cheetah Run | $0.643 \pm 0.131$ | $0.689 \pm 0.113$ | $0.695 \pm 0.087$ | $\mathbf{0.725 \pm 0.049}$ |
| Cup Catch | $0.652 \pm 0.087$ | $0.701 \pm 0.072$ | $0.714 \pm 0.061$ | $\mathbf{0.728 \pm 0.021}$ |
| Walker Walk | $0.612 \pm 0.140$ | $0.696 \pm 0.063$ | $0.701 \pm 0.073$ | $\mathbf{0.730 \pm 0.034}$ |
| Quadruped Walk | $0.656 \pm 0.092$ | $0.701 \pm 0.067$ | $0.703 \pm 0.072$ | $\mathbf{0.723 \pm 0.039}$ |
| Hopper Hop | $0.633 \pm 0.127$ | $0.704 \pm 0.087$ | $0.701 \pm 0.092$ | $\mathbf{0.722 \pm 0.038}$ |

**Trial efficiency.** Figure 4 presents test returns under limited environment trials. The number of environment trials serves as the x-axis to quantify exploration efficiency by measuring performance under the same environment exploration opportunities, which is a crucial factor for real-world applications and high-cost simulations. The complete results for 20 DMC tasks under limited environment trials are provided in Appendix H.2. The results show that the proposed DMWM-AC and DMWM-MPC significantly outperform baseline methods across most tasks with limited

environment trials, particularly in complex environments such as Cheetah Run and Finger Turn Hard, and Quadruped Run. In contrast, Dreamer and GD-MPC exhibit limited performance in high-dimensional control tasks with lower learning efficiency and stability. As shown in Figure 4, DMWM approaches achieve an average 5.5-fold improvement in test return under limited environment trials compared to baseline methods. These observations highlight that leveraging logical information from the environment enhances exploration efficiency, especially in complex tasks, by enhancing DMWM's capability in long-term imagination.

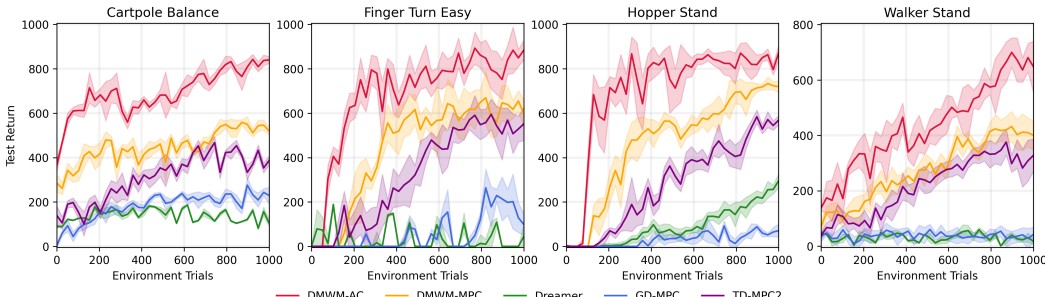

Figure 4: Performance comparison of results on 4 DMC tasks under environment trials that indicate the number of times that models explore the environments. The vertical axis indicates the average return over 100 test episodes. Complete results on 20 DMC tasks are concluded in Appendix H.2.

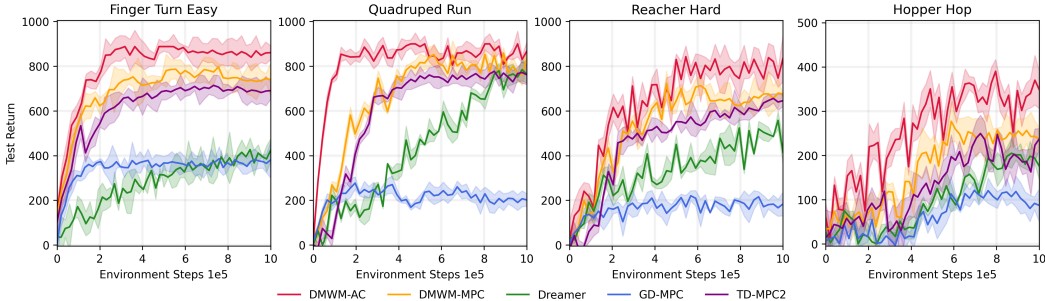

Figure 5: Performance comparison on 4 DMC tasks under environment steps that indicate the number of environment interactions. The vertical axis denotes the average test return over 100 episodes. Complete test results on 20 DMC tasks are provided in Appendix H.3.

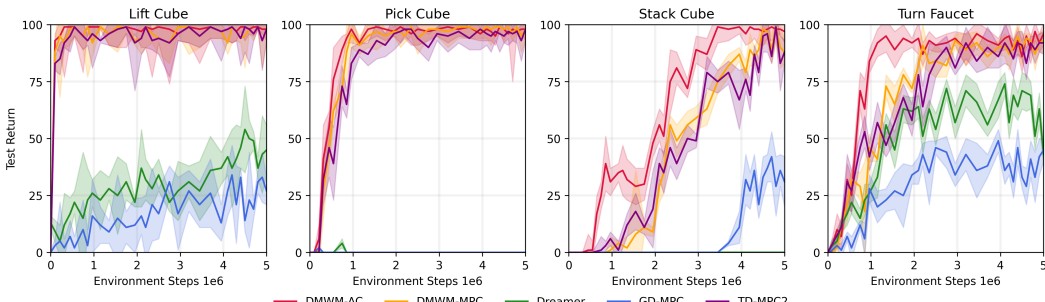

Figure 6: Performance comparison of test results on 4 ManiSkill2 robotic tasks under environment steps. The vertical axis denotes the average test return over 100 episodes.

**Data efficiency.** Figures 5-7 present test results under limited environment steps for the purpose of quantifying the data efficiency. The complete results for 20 DMC tasks under limited environment steps are provided in Appendix H.3. DMWM approaches consistently outperform Dreamer and GD-MPC across most tasks in terms of convergence speed and final returns, particularly in high-dimensional dynamics such as Cheetah Run and Quadruped Run. These results highlight DMWM's superior long-term planning and dynamic modeling capabilities, enabling more efficient utilization of environment data. For tasks that require simple dynamic modeling like Cartpole Balance and Cup Catch, all methods converge rapidly. In contrast, for complex tasks, such as Reacher Hard and Hopper Hop, the proposed DMWM approaches achieve stable performance and an average improvement of 32% in test return under limited data compared to Dreamer and GD-MPC.

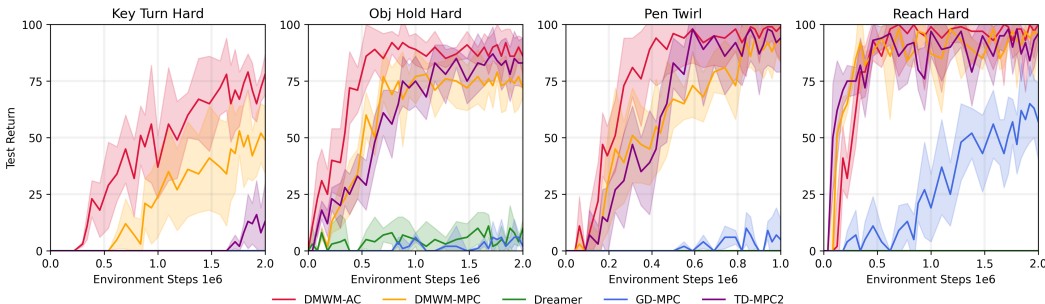

Figure 7: Performance comparison of test results on 4 MyoSuite robotic tasks under environment steps. The vertical axis denotes the average test return over 100 episodes.

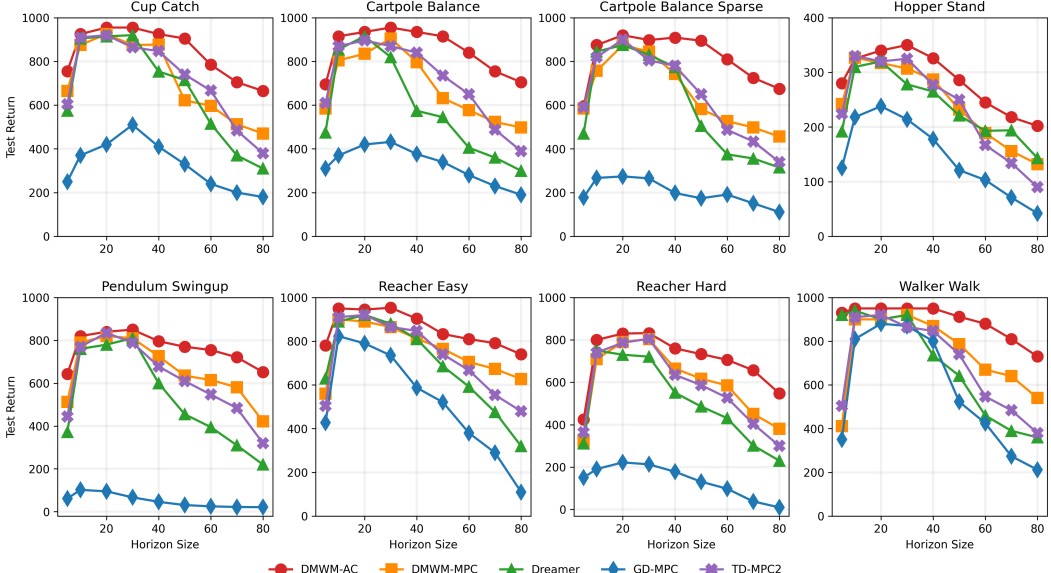

Figure 8: Performance comparison on 8 DMC tasks across different horizon size. Complete results on 20 DMC tasks across varying horizon size are provided in Appendix H.4.

**Imagination ability over extended horizon size.** As shown in Figure 8, although a long-horizon prediction introduces cumulative errors, DMWM-AC and DMWM-MPC consistently achieve high test returns across most tasks and remain stable performance even over extended horizons. In stability-critical tasks, such as Cartpole Balance and Pendulum Swingup, DMWM approaches maintain strong performance across a wide range of prediction horizons, whereas Dreamer and GD-MPC are more susceptible to degradation due to prediction errors. For extended horizon size of $H > 30$ in complex control tasks, DMWM approaches achieve an average 120% improvement in test return compared to Dreamer and GD-MPC. These results emphasize the crucial role of logical reasoning in long-term imagination. Across most tasks, DMWM approaches demonstrate superior performance over long-term horizons.

**Impact of logic inference depth.** Figure 9 presents test returns with inference depth $\alpha = 10, 30, 50$ and the complete results for 20 DMC tasks are provided in Appendix H.5. From Figure 9, we observe that a bigger reasoning depth $\alpha$ can produce performance gains in long-horizon decision making along with diminishing marginal returns from increasing logical inference depth and increased computational overhead. Moreover, Figure 9 also shows that the proposed deep logic inference can effectively capture the logic over the long-horizon trajectories for more robust imagination.

## 4 Related Works

World models are critical components in model-based intelligent systems, which enable learning, reasoning and decision-making in complex environments [5, 6, 7, 32, 33, 34, 35, 36, 37]. By

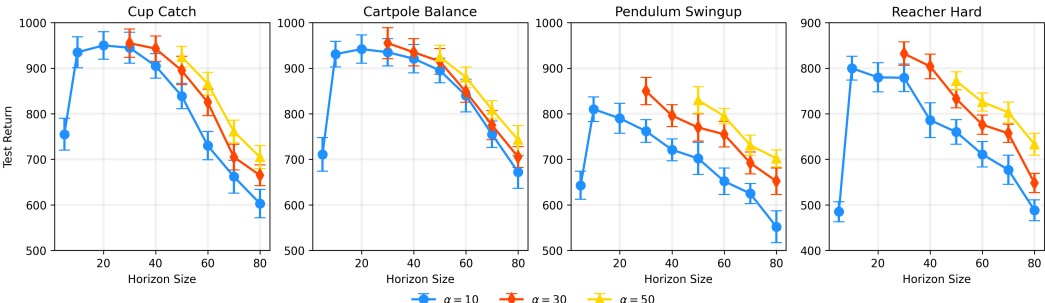

Figure 9: Performance comparison of results on 4 DMC tasks under environment trials that indicate the number of times that models explore the environments. The vertical axis indicates the average return over 100 test episodes. Complete results on 20 DMC tasks are concluded in Appendix H.5.

simulating the environment, these models imagine future states, plan behaviors and optimize strategies without relying heavily on real-world trial and error. Existing frameworks for world modeling, such as RSSM [5, 6, 7], generative models [32, 33, 34], and large language models (LLM) [35, 36, 37], have demonstrated varying strengths in tasks, such as prediction and efficient behavior planning. The Dreamer series [5, 6, 7], for instance, has advanced model-based reinforcement learning by improving task generalization and sample efficiency through latent space modeling and planning. Additionally, diffusion model-based world models [32, 33, 34] can generate diverse and high-quality outputs but cannot ensure long-term logical consistency. On the other hand, LLM-based models [35, 36, 37] can perform limited logical reasoning and task decomposition but face challenges in dynamic environment modeling and resource consumption.

Furthermore, logic neural reasoning is a promising approach to enhance the generalization ability and long-term imagination of world models [38, 29, 39]. Previous research has explored logic neural networks and probabilistic logic [40, 30, 41], but these methods have struggled with dynamic environments and evolving logical variables. Our approach, by contrast, enables the automatical and implicit logic inference, thus providing superior generalization and robustness in complex, dynamic task settings. Hence, in contrast to the existing world models, our work introduces a novel dual-mind framework that combines the efficient sampling of RSSM with logic-driven reasoning to enhance long-term imagination with logical consistency, thus addressing a critical research gap for logic-driven robust world models.

## 5    Conclusion and Limitations

**Conclusion.**   In this paper, we have proposed DMWM, a novel world model framework for reliable long-term imagination inspired by the dual-process theory of human cognition. The proposed DMWM combines the fast, intuitive-driven S1 with the structured, logic inference-driven S2. We have designed an efficient inter-system feedback mechanism that enhances the logic consistency and adaptability of imagination trajectories. Extensive evaluations on DMControl and robotic tasks across diverse benchmarks demonstrated significant improvements in logical consistency, data-efficiency and reliable imagination over an extended horizon size.

**Limitations and future work.**   A key limitation of our approach lies in its reliance on predefined simple domain-specific logical rules, which restricts its adaptability to environments where such rules are ambiguous or constantly evolving. This dependency limits the framework's ability to generalize to novel tasks. Future work could focus on enabling the model to autonomously learn and adapt logical rules from data by exploring causal relationships. This, in turn, can reduce the need for explicit manual definitions and enhance flexibility in dynamic and complex environments.

## Acknowledgments

This work was supported by the US National Science Foundation under Grants CNS-2225511, IIS-2509636, DBI-2412389, CMMI-2240402, IIS-2312794, and CCF-1918770. Any opinions, findings, and conclusions or recommendations expressed in this material are those of the author(s) and do not necessarily reflect the views of the sponsor(s).

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

**Contents of Appendix**

# A    Impact Statement

Inspired by the human cognitive model, our work proposes a novel DMWM architecture that integrates intuitive RSSM-S1 with logic-driven LINN-S2 for the first time, which addresses long-horizon imagination for model-based RL and MPC. By enhancing logical consistency of the architecture, DMWM improves both efficiency and robustness in complex tasks. Because of these benefits, DMWM provides a practical and interpretable solution for real-world applications such as autonomous driving and robotic planning.

Moreover, this work presents a generalized dual-mind framework for world models that can serve as a solid foundation for future research on general world models. We believe this approach marks a meaningful step toward artificial general intelligence (AGI) by bridging intuitive processes with logical reasoning. Hence, DWMW paves the way for more robust, logical and human-like decision-making systems.

# B    Background

## B.1    Actor-Critic

The actor-critic model is a widely used algorithmic framework in Reinforcement Learning (RL), which combines the advantages of policy gradient methods and value function approximation. It addresses RL tasks through the collaborative learning of two primary components: the Actor (policy network) and the Critic (value network), given as follows:

$$\begin{aligned} \text{Action model:} \quad & a_\tau \sim q_\vartheta(a_\tau \mid s_\tau) \\ \text{Value model:} \quad & v_\psi(s_\tau) \approx \mathbb{E}_{q(\cdot|s_\tau)}\left(\sum_{t=\tau}^{H} \gamma^{t-\tau} r_\tau\right). \end{aligned} \tag{18}$$

In the world Model, the target of the actor-critic model is to maximize the reward over the imagined trajectories with the horizon size $H$. Specifically, the action model seeks to maximize an estimated value, while the value model strives to accurately predict the value estimate, which evolves as the action model updates. The training target of the Actor and the Value are given as follows [58, 5, 11].

$$\vartheta^* = \max_{\vartheta} \mathbb{E}_{q_\phi, q_\vartheta}\left[\sum_{\tau=t}^{t+H} V_\lambda(s_\tau)\right] \tag{19}$$

$$\psi^* = \min_{\psi} \mathbb{E}_{q_\phi, q_\vartheta}\left[\sum_{\tau=t}^{t+H} \frac{1}{2}\left(v_\psi(s_\tau) - V_\lambda(s_\tau)\right)^2\right] \tag{20}$$

$$V_\lambda(s_\tau) = (1-\lambda)\left(\sum_{n=1}^{H-1} \lambda^{n-1} V_n^N(s_\tau)\right) + \lambda^{H-1} V_H^N(s_\tau) \tag{21}$$

$$V_k^N(s_\tau) = \mathbb{E}_{q_\phi, q_\vartheta}\left[\sum_{n=\tau}^{h-1} \gamma^{n-\tau} r_n + \gamma^{h-\tau} v_\psi(s_h)\right], \tag{22}$$

where $h = \min(\tau + k, t + H)$, $\vartheta$ represents the parameters of the Actor $\psi$ represents the parameters of the Critic, and $\lambda$ is the discount factor.

## B.2    Model Predictive Control

Model predictive control (MPC) is an optimization-based control method extensively applied in engineering and industrial control systems [13, 59, 60, 61]. By leveraging the system's dynamic model, MPC predicts future behavior through rolling optimization, generating optimal control inputs at each time step to achieve the desired system objectives. However, due to the fact that MPC strongly relies on the system model and the requirement for online optimization, it has difficulty in effective decision-making performance if the world model fails to provide stable and reliable

imagined trajectories [9, 11]. The gradient-based MPC framework [11] for world model is presented as

$$a_{t:t+H}^* = \max_{a_{t:t+H}^{(j)}} R^{(j)}, \quad R^{(j)} = \sum_{\tau=t+1}^{t+H+1} \mathbb{E}\big[q_\phi(r_\tau \mid s_\tau^{(j)})\big] \tag{23}$$

$$\left\{a_{t:t+H}^{(j)}\right\}_{j=1}^J \sim \mathcal{N}(\mu_t, \mathrm{diag}(\sigma_t^2)) \tag{24}$$

$$s_{t:t+H+1}^{(j)} \sim q_\phi(s_t \mid o_{1:t}^{(j)}, a_{1:t-1}^{(j)}) \prod_{\tau=t+1}^{t+H+1} p_\phi(s_\tau \mid s_{\tau-1}^{(j)}, a_{\tau-1}^{(j)}) \tag{25}$$

$$a_{t:t+H}^{(j)} = a_{t:t+H}^{(j)} - \nabla R^{(j)}, \tag{26}$$

where $a_{t:t+H}^{(j)}$ and $s_{t:t+H}^{(j)}$ are sequences of states and actions of the candidate $j$ from the timestep $t$ to the timestep $t + H$.

## C   Derivation of Logical ELBO for Observation Loss

The ELBO with logic rules for the observation loss can be derived by

$$
\begin{aligned}
\ln p_\varphi\left(o_{1:T} \mid a_{1:T}\right) &= \int p_\varphi(o_{1:T}, z_{1:T} \mid a_{1:T}) \, dz_{1:T} \\
&= \mathbb{E}_{q_\varphi(z_{1:T}|o_{1:T},a_{1:T})}\left[\frac{p_\varphi(o_{1:T}, z_{1:T} \mid a_{1:T})}{q_\varphi(z_{1:T} \mid o_{1:T}, a_{1:T})}\right] \\
&= \ln \mathbb{E}_{q_\varphi(z_{1:T}|o_{1:T},a_{1:T})}\left[\prod_{t=1}^T \frac{p_\varphi(o_t \mid z_t)p_\varphi(z_t \mid z_{t-1}, a_{t-1})\mathcal{C}(\phi(z_t, z_{t-1}, a_{t-1}))}{q_\varphi(z_t \mid o_{\leq t}, a_{\leq t})}\right] \\
&\geq \mathbb{E}_{q_\varphi(z_{1:T}|o_{1:T},a_{1:T})}\left[\sum_{t=1}^T \ln p_\varphi(o_t \mid z_t) + \ln p_\varphi(z_t \mid z_{t-1}, a_{t-1})\right. \\
&\qquad\qquad\qquad\qquad \left. + \ln \mathcal{C}(\phi(z_t, z_{t-1}, a_{t-1}) - \ln q_\varphi(z_t \mid o_{\leq t}, a_{\leq t}))\right] \\
&= \sum_{t=1}^T \left(\underbrace{\mathbb{E}_{q_\varphi(z_t|o_{\leq t},a_{<t})}\left[\ln p_\varphi\left(o_t \mid z_t\right)\right]}_{\text{Decoding}} + \underbrace{\mathbb{E}_{q_\varphi(z_t|o_{\leq t},a_{<t})}\left[\ln \mathcal{C}(\phi(z_t, z_{t-1}, a_{t-1}))\right]}_{\text{Logic Inference}}\right. \\
&\qquad\qquad \left. - \underbrace{\mathbb{E}_{q_\varphi(z_{t-1}|o_{\leq t-1},a_{<t-1})}\left[\mathrm{KL}\left[q_\varphi\left(z_t \mid o_{\leq t}, a_{<t}\right) \| p_\varphi\left(z_t \mid z_{t-1}, a_{t-1}\right)\right]\right]}_{\text{Prediction}}\right),
\end{aligned}
\tag{27}
$$

where $\mathcal{C}(\phi(z_t, z_{t-1}, a_{t-1})) = \mathrm{Sim}(\phi(z_t, z_{t-1}, a_{t-1}), \mathbf{T})$ is the logical consistency function that measures whether the logical rule $\phi : z_{t-1} \wedge a_{t-1} \to z_t$ is satisfied by imagination.

— Appendices continue on next page —

# D Algorithms

## D.1 DMWM With Actor-Critic

The training process of DMWM with actor-critic-based decision module is shown in Algorithm 1.

---

**Algorithm 1** DMWM With Actor-Critic

---

Hyper Parameters: seed episode $S$, training episodes $N$, batch size $B$, collect interval $C$,
                      sequence length $L$, imagination horizon $H$, learning rate $\eta$.
Initialize dataset $\mathcal{D}$ with $S$ seed episodes.
Initialize DMWM parameters $\phi, w$.
Initialize Actor-Critic parameters $\vartheta, \psi$.
**for** Training step $n \rightarrow N$ **do**
    **for** Collect interval $c \rightarrow C$ **do**
        // System 1 Training
        Sample $B$ Sequences $\{(o_t, a_t, r_t)\}_{t=k}^{k+L} \sim \mathcal{D}$.
        Compute $h_t = f_\varphi(h_{t-1}, z_{t-1}, a_{t-1})$.
        Predict $\hat{z}_t \sim p_\varphi(\hat{z}_t \mid h_t)$, $\hat{o}_t \sim p_\varphi(\hat{o}_t \mid h_t, z_t)$.
        Update S1 $\psi \leftarrow \psi - \eta_\psi \nabla_\psi \mathcal{L}_{\text{S1}}(\psi)$.
        // System 2 Training
        Learn Logic Regularizations $\mathcal{L}_{\text{reg}}$.
        // S1's Guidance on S2 Based on Truth $\{(h_t, a_t, h_{t+1})\}$
        Compute $L_T^\lambda = \phi_1^\lambda \wedge \phi_2^\lambda \wedge \phi_3^\lambda \cdots \phi_{T-1}^\lambda$.
        Update S2 $w \leftarrow w - \eta_w \nabla_w \mathcal{L}_{\text{S1}}(w)$.
        // Actor-Critic Training
        Imagine $\{(z_\tau, a_\tau)\}_{\tau=t}^{t+H}$ from each $z_t$.
        Predict rewards $\mathbb{E}(q_\varphi(r_\tau \mid h_\tau, z_\tau))$ and values $v_\psi(s_\tau)$.
        Compute value estimates $V_\lambda(s_\tau)$.
        Update $\vartheta \leftarrow \vartheta + \eta_\vartheta \nabla_\vartheta \sum_{\tau=t}^{t+H} V_\lambda(s_\tau)$.
        Update $\psi \leftarrow \psi - \eta_\psi \nabla_\psi \sum_{\tau=t}^{t+H} \frac{1}{2}\|v_\psi(s_\tau) - V_\lambda(s_\tau)\|^2$.
        // S2's Guidance on S1 Based on Imagination $\{(z_\tau, a_\tau)\}$
        Compute Logic Consistency of $\{(z_\tau, a_\tau)\}$ With $L_T^\lambda$
        Update S1 $\psi \leftarrow \psi - \eta_\psi \nabla_\psi \mathcal{L}_{\text{S2}}(\psi)$.
    **end for**
    // Real Environment Interaction & Data Collection
    Start a environment $env.reset()$
    **for** Time step $t \rightarrow T$ **do**
        Compute $h_t = f_\varphi(h_{t-1}, z_{t-1}, a_{t-1})$.
        Compute $z_t \sim q_\varphi(z_t \mid h_t, o_t)$
        Obtain $a_t \sim q_\vartheta(a_t \mid z_t)$ from decision-making model.
        Interact $r_t, o_{t+1} \leftarrow env.step(a_t)$.
    **end for**
    Add experience to dataset $\mathcal{D} \leftarrow \mathcal{D} \cup \{(o_t, a_t, r_t)\}_{t=1}^T$.
**end for**

---

## D.2 DMWM With MPC

The training process of DMWM with Grad-MPC for decision-making [11] is shown in Algorithm 2.

---

**Algorithm 2** DMWM With Grad-MPC

---

Hyper Parameters: iterations $I$, candidate Size $J$, learning rate $\eta_R$.
Initialize dataset $\mathcal{D}$ with $S$ seed episodes.
Initialize DMWM parameters $\phi, w$.
**for** Training step $n \to N$ **do**
  $\cdots$
  // Real Environment Interaction & Data Collection
  Start a environment $env.reset()$
  **for** Time step $t \to T$ **do**
    Compute $h_t = f_\varphi(h_{t-1}, z_{t-1}, a_{t-1})$.
    Compute $z_t \sim q_\varphi(z_t \mid h_t, o_t)$
    // MPC Decision Making
    Sample Actions $\left\{ a_{t:t+H}^{(j)} \right\}_{j=1}^{J} \sim \mathcal{N}(\mu_t, \mathrm{diag}(\sigma_t^2))$.
    **for** Iteration $i \to I$ **do**
      **for** Candidate sequence $j \to J$ **do**
        $s_{t:t+H+1}^{(j)} \sim q_\phi(s_t \mid o_{1:t}^{(j)}, a_{1:t-1}^{(j)}) \prod_{\tau=t+1}^{t+H+1} p_\phi(s_\tau \mid s_{\tau-1}^{(j)}, a_{\tau-1}^{(j)})$.
        $R^{(j)} = \sum_{\tau=t+1}^{t+H+1} \mathbb{E}\left[ p(r_\tau \mid s_\tau^{(j)}) \right]$.
        Update Action $a_{t:t+H}^{(j)} = a_{t:t+H}^{(j)} - \eta_R \nabla R^{(j)}$.
      **end for**
    **end for**
    $a_{t:t+H}^* = \max_{a_{t:t+H}^{(j)}} R^{(j)}$
    Interact $r_t, o_{t+1} \leftarrow env.step(a_t^*)$.
  **end for**
  Add experience to dataset $\mathcal{D} \leftarrow \mathcal{D} \cup \{(o_t, a_t, r_t)\}_{t=1}^{T}$.
  $\cdots$
**end for**

---

# E Hyperparameters

## E.1 Environment Setting

The action repeat setting for different DMControl tasks and robotic tasks is presented in TABLE 2.

Table 3: Action Repeat Setting and Action Dim

| Env | Task | Action Dim | Action Repeat |
|---|---|---|---|
| DMC | Cartpole Swingup | 1 | 8 |
| | Pendulum Swingup | 1 | 6 |
| | Reacher Easy | 2 | 4 |
| | Finger Spin | 2 | 2 |
| | Cheetah Run | 6 | 4 |
| | Cup Catch | 2 | 6 |
| | Walker Walk | 6 | 2 |
| | Quadruped Walk | 12 | 2 |
| | Hopper Hop | 4 | 2 |
| MyoSuite | Key Turn Hard | 39 | 1 |
| | Object Hold Hard | 39 | 1 |
| | Pen Twirl | 39 | 1 |
| | Reach Hard | 39 | 1 |
| ManiSkill2 | Lift Cube | 4 | 2 |
| | Pick Cube | 4 | 2 |
| | Stack Cube | 4 | 2 |
| | Turn Faucet | 7 | 2 |

## E.2 Model Setting

The hyperparameters of models are presented in TABLE 3.

Table 4: Hyperparameter Setting

| Parameter | Symbol | Value |
|---|---|---|
| **Dual-Mind World Model (General)** | | |
| Replay memory size | — | 1e6 |
| Batch size | $B$ | 50 |
| Sequence length | $L$ | 64 |
| Seed episode | $S$ | 5 |
| Training episodes | $N$ | 1e3 |
| Collect Interval | $C$ | 100 |
| Max episode length | — | 500 |
| Exploration noise | — | 0.3 |
| Imagination horizon | $H$ | 30 |
| Gradient clipping | — | 100 |
| **RSSM-S1** | | |
| Activation function | — | Relu |
| Embedding size | — | 1024 |
| Hidden size | — | 200 |
| Belief size | — | 200 |
| State size | — | 30 |
| Overshooting distance | — | 50 |
| Overshooting KL-beta | — | 0 |
| Global KL-beta | — | 0 |
| overshooting reward scale | — | 0 |
| Free nats | — | 3 |
| Bit-depth | — | 5 |
| Weights | $\varpi_{\mathrm{dyn}}, \varpi_{\mathrm{rep}}$ | 1 |
| Optimizer | — | Adam |
| Adam epsilon | — | 1e-4 |
| Learning rate | $\eta_\psi$ | 1e-3 |
| **LINN-S2** | | |
| Reasoning depth | $\alpha$ | 30 |
| Logic vector size | $|v|, |m|$ | 64 |
| L2 weight | $\beta_{\ell_2}$ | 1e-5 |
| Regularization weight | $\beta_{\mathrm{reg}}$ | 1 |
| Logic MLP number | — | 3 |
| Optimizer | — | SGD |
| Learning rate | $\eta_w$ | 1e-2 |
| **Actor-Critic** [5] | | |
| Return lambda | $\lambda$ | 0.95 |
| Planning horizon discount | — | 0.99 |
| Optimizer | — | Adam |
| Adam epsilon | — | 1e-4 |
| Learning rate | $\eta_\vartheta, \eta_\psi$ | 1e-4 |
| **Grad-MPC** [11] | | |
| Iterations | $I$ | 40 |
| Candidate Size | $J$ | 1000 |
| Learning Rate | $\eta_R$ | 0.1-0.01-0.005-0.0001 |
| **TD-MPC2** (refer to [13]) | | |

# F  Further Related Work

## F.1  World Model

World models are fundamental building blocks for model-based intelligent systems to make decisions, learn, and reason in complex environments. It enables prediction, efficient behavior planning through environment simulation, and strategy optimization using virtual simulations, thus reducing reliance on real-world trial and error [5, 6, 7, 62, 63]. Existing world modeling frameworks can be categorized as recurrent state space models (RSSMs) [2, 5, 6, 7, 10, 64, 65, 13, 66, 3, 15, 67], generative models [32, 33, 68, 69, 70, 34, 71, 72, 12, 73], and large language model (LLM) [35, 36, 74, 75, 37, 76].

The Dreamer series [5, 6, 7] lays an important foundation for general model-based reinforcement learning (MBRL) by continuously improving sample efficiency and task generalization ability through dynamic modeling and planning in the latent space. To adapt the RSSM-based world model for real-world environments, [2] optimized confidence and entropy regularization for the gradient to mitigate discrepancies between virtual simulations and real-world environments. [1] studied an object-centric world model, and introduced an object-state recurrent neural network (OS-RNN) to follow the object states. [12] extended multimodal RSSM to support joint text and visual inputs. [7, 13, 67] explored the multiple tasks with RSSM-based world models. [13] utilized SimNorm to sparse and normalize the potential states, which projected the latent representations to simplices of fixed dimensions, thus mitigating the gradient explosion and improving the training stability. [15] introduced masking-based latent reconstruction with a dual branch structure to handle the exogenous noise in the complex environment. [3] introduced the hierarchical imagination with parallel processing and proposed efficient time-balanced sampling.

The diffusion model-based world model offers high-quality and diverse generations with temporal smoothness and consistency, achieved through denoising and time inversion processes [32, 33, 68, 69, 70, 34, 12]. However, its reliance on multi-step denoising significantly slows downsampling, inference, and computation compared to RSSM, making it less suitable for real-time tasks and challenging to maintain long-term consistency [68, 69, 70, 12]. Additionally, unlike RSSM's latent space modeling, diffusion models cannot extract task-relevant features and accurately capture environment dynamics, particularly in complex environments with long-tailed distributions [32, 68, 70, 71, 12]. The LLM-based world model serves as a powerful tool for complex task planning and execution through its logical reasoning capability and dynamic controllability [35, 36, 74], and is able to realize task decomposition and cross-domain knowledge transfer through natural language instructions [76]. However, the model cannot perform in dynamic modeling, and the generated states and actions often cannot accurately reflect the dynamic changes in the environment. In addition, its high reliance on linguistic representation may lead to insufficient quality of modal alignment, posing the risk of information loss or misinterpretation, while the significant consumption of computational resources during training and inference limits the application of LLM-based world models in resource-constrained environments [35, 36, 74, 75].

Unlike all of the aforementioned works, we focus on the long-term imagination of world models and proposes to enable the logical consistency of state representations and predictions over an extended horizon for the first time. We retain the efficient sampling and representation capabilities of RSSM as System 1 and integrate a logic-integrated neural network (LINN) as System 2 to imbue the world model with logic reasoning capabilities. The proposed DMWM thus delivers reliable and efficient long-term imagination with logical consistency, addressing a critical research gap in existing approaches.

## F.2  Logic Neural Reasoning

Logic neural reasoning enhances the logical consistency, long-term imagination, and generalization ability of world models by embedding logical rules and reasoning capabilities [38, 29]. By combining the interpretability of symbolic logic with the expressive power of neural networks, it provides robust and rapid reasoning for complex task planning [39, 40, 30, 41, 31, 77, 78]. However, explicitly modeling the logical structure of the world presents significant challenges due to the inherent ambiguity of logical rules, the prevalence of noisy data, and the dynamic and complex nature of logical interactions. [39] converted first-order logic rules into computational graphs for neural networks, enhancing their learning capabilities in low-data environments while providing limited support for complex temporal logic. [40] integrated Markov logic networks with knowledge graph

embeddings to address uncertainty in logical reasoning. In a logical neural network (LNN) [41], neurons are mapped neurons to logical formulas, enabling each neuron to function as a logical operator. However, LNNs rely on predefined and static logical rules, facing limitations in generalization when confronted with uncertainty or evolving logical variables. The work in [78] introduced inductive logic programming and automatically learned logic rules. However, the aforementioned approaches make it hard to capture the environment dynamics and extend the long-term imagination capability of the world model. In contrast, the concept of a LINN [30, 31] introduces a paradigm shift by dynamically constructing computational graphs and employing neural modules to learn logical operations. Thus, LINN can automatically infer implicit logical rules, circumventing the need for explicit specification. By integrating neural flexibility with logical rigor [30, 31], LINN achieves superior generalization capacity and enhanced robustness to noise, making it well-suited for reasoning tasks in dynamic and complex environments.

In this work, we enhance the long-term imagination capabilities of world models through logical reasoning, a critical yet unexplored area in world model research, with significant implications for creating advanced and general artificial intelligence (AGI).

# G    Complete Logic Regularization and Rules Table

The complete logic rules and the corresponding regularizers are given in TABLE 5 [30] for negation $\neg$, conjunction $\wedge$, disjunction $\vee$ and implication $\rightarrow$.

Table 5: Complete Logical Regularizers and Rules for System 2

| | Logical Rule | Equation | Logic Regularizer $r_i$ |
|---|---|---|---|
| $\neg$ | **Negation** | $w \rightarrow \mathbf{T} = \mathbf{T}$ | $r_1 = \sum_{w \in W \cup \{\mathbf{T}\}} \mathrm{Sim}(\mathrm{NOT}(w), w)$ |
| | **Negations** | $\neg(\neg w) = w$ | $r_2 = \sum_{w \in W} 1 - \mathrm{Sim}(\mathrm{NOT}(\mathrm{NOT}(w)), w)$ |
| $\wedge$ | **Identity** | $w \wedge \mathbf{T} = w$ | $r_3 = \sum_{w \in W} 1 - \mathrm{Sim}(\mathrm{AND}(w, \mathbf{T}), w)$ |
| | **Annihilator** | $w \wedge \mathbf{T} = w$ | $r_4 = \sum_{w \in W} 1 - \mathrm{Sim}(\mathrm{AND}(w, \mathbf{F}), \mathbf{F})$ |
| | **Idempotence** | $w \wedge \mathbf{F} = \mathbf{F}$ | $r_5 = \sum_{w \in W} 1 - \mathrm{Sim}(\mathrm{AND}(w, w), w)$ |
| | **Complement** | $w \wedge w = w$ | $r_6 = \sum_{w \in W} 1 - \mathrm{Sim}(\mathrm{AND}(w, \mathrm{NOT}(w)), \mathbf{F})$ |
| $\vee$ | **Identity** | $w \vee \mathbf{F} = w$ | $r_7 = \sum_{w \in W} 1 - \mathrm{Sim}(\mathrm{OR}(w, \mathbf{F}), w)$ |
| | **Annihilator** | $w \vee \mathbf{T} = \mathbf{T}$ | $r_8 = \sum_{w \in W} 1 - \mathrm{Sim}(\mathrm{OR}(w, \mathbf{T}), \mathbf{T})$ |
| | **Idempotence** | $w \vee w = w$ | $r_9 = \sum_{w \in W} 1 - \mathrm{Sim}(\mathrm{OR}(w, w), w)$ |
| | **Complement** | $w \vee \neg w = \mathbf{T}$ | $r_{10} = \sum_{w \in W} 1 - \mathrm{Sim}(\mathrm{OR}(w, \mathrm{NOT}(w)), \mathbf{T})$ |
| $\rightarrow$ | **Identity** | $w \rightarrow \mathbf{T} = \mathbf{T}$ | $r_{11} = \sum_{w \in W} 1 - \mathrm{Sim}(\mathrm{OR}(\mathrm{NOT}(w), \mathbf{T}), \mathbf{T})$ |
| | **Annihilator** | $w \rightarrow \mathbf{F} = \neg w$ | $r_{12} = \sum_{w \in W} 1 - \mathrm{Sim}(\mathrm{OR}(\mathrm{NOT}(w), \mathbf{F}), \mathrm{NOT}(w))$ |
| | **Idempotence** | $w \rightarrow w = \mathbf{T}$ | $r_{13} = \sum_{w \in W} 1 - \mathrm{Sim}(\mathrm{OR}(\mathrm{NOT}(w), w), \mathbf{T})$ |
| | **Complement** | $w \rightarrow \neg w \equiv \neg w$ | $r_{14} = \sum_{w \in W} 1 - \mathrm{Sim}(\mathrm{OR}(\mathrm{NOT}(w), \mathrm{NOT}(w)), \mathrm{NOT}(w))$ |

# H    Additional Experiments

## H.1    Logical Consistency

Table 6: Logical consistency comparison between our proposed DMWM and various RSSM baselines across 20 DMC tasks. We report the mean and variance of logical consistency from imagination over 100 test episodes with the horizon size of $H = 10, 30, 50, 100$.

| Env | $H$ | Dreamer | Hieros | HRSSM | DMWM (**Ours**) |
|---|---|---|---|---|---|
| Acrobot Swingup | 10 | $0.713 \pm 0.031$ | $0.730 \pm 0.012$ | $0.722 \pm 0.017$ | $0.733 \pm 0.007$ |
| | 30 | $0.667 \pm 0.063$ | $0.704 \pm 0.032$ | $0.712 \pm 0.044$ | $\mathbf{0.731 \pm 0.017}$ |
| | 50 | $0.568 \pm 0.129$ | $0.692 \pm 0.058$ | $0.687 \pm 0.083$ | $0.715 \pm 0.032$ |
| | 100 | $0.485 \pm 0.167$ | $0.672 \pm 0.112$ | $0.651 \pm 0.132$ | $0.699 \pm 0.078$ |
| Cartpole Balance | 10 | $0.721 \pm 0.023$ | $0.730 \pm 0.009$ | $0.729 \pm 0.011$ | $0.730 \pm 0.008$ |
| | 30 | $0.683 \pm 0.057$ | $0.711 \pm 0.032$ | $0.713 \pm 0.041$ | $\mathbf{0.727 \pm 0.023}$ |
| | 50 | $0.574 \pm 0.112$ | $0.687 \pm 0.084$ | $0.695 \pm 0.072$ | $0.717 \pm 0.045$ |
| | 100 | $0.491 \pm 0.171$ | $0.663 \pm 0.142$ | $0.655 \pm 0.121$ | $0.701 \pm 0.092$ |
| Cartpole Balance Sparse | 10 | $0.705 \pm 0.132$ | $0.719 \pm 0.032$ | $0.722 \pm 0.034$ | $0.722 \pm 0.026$ |
| | 30 | $0.602 \pm 0.167$ | $0.682 \pm 0.101$ | $0.693 \pm \mathbf{0.081}$ | $0.695 \pm 0.093$ |
| | 50 | $0.432 \pm 0.203$ | $0.632 \pm 0.178$ | $0.643 \pm 0.162$ | $0.652 \pm 0.135$ |
| | 100 | $0.321 \pm 0.286$ | $0.571 \pm 0.232$ | $0.589 \pm 0.213$ | $0.603 \pm 0.182$ |
| Cartpole Swingup | 10 | $0.719 \pm 0.031$ | $0.729 \pm 0.011$ | $0.723 \pm 0.023$ | $0.730 \pm 0.011$ |
| | 30 | $0.678 \pm 0.062$ | $0.698 \pm 0.043$ | $0.703 \pm 0.082$ | $0.723 \pm 0.032$ |
| | 50 | $0.563 \pm 0.091$ | $0.672 \pm 0.124$ | $0.662 \pm 0.145$ | $0.702 \pm 0.078$ |
| | 100 | $0.474 \pm 0.158$ | $0.621 \pm 0.191$ | $0.602 \pm 0.191$ | $0.672 \pm 0.152$ |
| Cartpole Swingup Sparse | 10 | $0.703 \pm 0.142$ | $0.717 \pm 0.036$ | $0.715 \pm 0.052$ | $0.720 \pm 0.030$ |
| | 30 | $0.613 \pm 0.187$ | $0.669 \pm 0.121$ | $0.671 \pm 0.098$ | $0.699 \pm 0.087$ |
| | 50 | $0.443 \pm 0.213$ | $0.621 \pm 0.185$ | $0.621 \pm 0.173$ | $0.669 \pm 0.121$ |
| | 100 | $0.403 \pm 0.252$ | $0.532 \pm 0.241$ | $0.559 \pm 0.231$ | $0.627 \pm 0.162$ |
| Cheetah Run | 10 | $0.709 \pm 0.085$ | $0.723 \pm 0.031$ | $0.721 \pm 0.023$ | $0.730 \pm 0.016$ |
| | 30 | $0.643 \pm 0.131$ | $0.689 \pm 0.113$ | $0.695 \pm 0.087$ | $0.725 \pm 0.049$ |
| | 50 | $0.527 \pm 0.165$ | $0.651 \pm 0.147$ | $0.667 \pm 0.131$ | $0.703 \pm 0.092$ |
| | 100 | $0.428 \pm 0.221$ | $0.606 \pm 0.186$ | $0.627 \pm 0.183$ | $0.676 \pm 0.142$ |
| Cup Catch | 10 | $0.711 \pm 0.049$ | $0.728 \pm 0.010$ | $0.726 \pm 0.014$ | $0.732 \pm 0.008$ |
| | 30 | $0.652 \pm 0.087$ | $0.701 \pm 0.072$ | $0.714 \pm 0.061$ | $\mathbf{0.728 \pm 0.021}$ |
| | 50 | $0.534 \pm 0.113$ | $0.681 \pm 0.098$ | $0.683 \pm 0.112$ | $0.712 \pm 0.053$ |
| | 100 | $0.465 \pm 0.181$ | $0.647 \pm 0.182$ | $0.633 \pm 0.172$ | $0.692 \pm 0.112$ |

| | | | | | |
|---|---|---|---|---|---|
| Finger Spin | 10 | $0.712 \pm 0.075$ | $0.727 \pm 0.011$ | $0.728 \pm 0.017$ | $0.733 \pm 0.009$ |
| | 30 | $0.647 \pm 0.102$ | $0.705 \pm 0.052$ | $0.712 \pm 0.057$ | $\mathbf{0.729 \pm 0.017}$ |
| | 50 | $0.517 \pm 0.131$ | $0.687 \pm 0.091$ | $0.679 \pm 0.136$ | $0.710 \pm 0.062$ |
| | 100 | $0.435 \pm 0.211$ | $0.636 \pm 0.165$ | $0.645 \pm 0.185$ | $0.684 \pm 0.127$ |
| Finger Turn Easy | 10 | $0.708 \pm 0.098$ | $0.726 \pm 0.013$ | $0.724 \pm 0.018$ | $0.732 \pm 0.015$ |
| | 30 | $0.634 \pm 0.113$ | $0.702 \pm 0.051$ | $0.705 \pm 0.067$ | $0.728 \pm 0.023$ |
| | 50 | $0.512 \pm 0.157$ | $0.681 \pm 0.117$ | $0.682 \pm 0.132$ | $0.712 \pm 0.067$ |
| | 100 | $0.412 \pm 0.191$ | $0.644 \pm 0.194$ | $0.617 \pm 0.197$ | $0.683 \pm 0.148$ |
| Finger Turn Hard | 10 | $0.704 \pm 0.118$ | $0.723 \pm 0.014$ | $0.724 \pm 0.021$ | $0.731 \pm 0.011$ |
| | 30 | $0.627 \pm 0.131$ | $0.698 \pm 0.061$ | $0.703 \pm 0.073$ | $0.725 \pm 0.029$ |
| | 50 | $0.487 \pm 0.162$ | $0.673 \pm 0.112$ | $0.673 \pm 0.152$ | $0.705 \pm 0.073$ |
| | 100 | $0.385 \pm 0.231$ | $0.623 \pm 0.201$ | $0.601 \pm 0.205$ | $0.675 \pm 0.142$ |
| Hopper Hop | 10 | $0.703 \pm 0.092$ | $0.729 \pm 0.023$ | $0.726 \pm 0.023$ | $0.731 \pm 0.017$ |
| | 30 | $0.633 \pm 0.127$ | $0.704 \pm 0.087$ | $0.701 \pm 0.092$ | $0.722 \pm 0.038$ |
| | 50 | $0.506 \pm 0.191$ | $0.664 \pm 0.132$ | $0.673 \pm 0.132$ | $0.698 \pm 0.092$ |
| | 100 | $0.407 \pm 0.237$ | $0.612 \pm 0.237$ | $0.623 \pm 0.201$ | $0.689 \pm 0.143$ |
| Hopper Stand | 10 | $0.709 \pm 0.056$ | $0.728 \pm 0.021$ | $0.725 \pm 0.019$ | $0.732 \pm 0.013$ |
| | 30 | $0.645 \pm 0.112$ | $0.699 \pm 0.057$ | $0.704 \pm 0.081$ | $0.724 \pm 0.039$ |
| | 50 | $0.523 \pm 0.168$ | $0.671 \pm 0.121$ | $0.689 \pm 0.137$ | $0.703 \pm 0.080$ |
| | 100 | $0.421 \pm 0.197$ | $0.632 \pm 0.211$ | $0.642 \pm 0.189$ | $0.692 \pm 0.139$ |
| Pendulum Swingup | 10 | $0.715 \pm 0.054$ | $0.728 \pm 0.017$ | $0.719 \pm 0.023$ | $0.732 \pm 0.015$ |
| | 30 | $0.611 \pm 0.137$ | $0.709 \pm 0.054$ | $0.699 \pm 0.079$ | $\mathbf{0.730 \pm 0.037}$ |
| | 50 | $0.526 \pm 0.161$ | $0.684 \pm 0.112$ | $0.664 \pm 0.146$ | $0.721 \pm 0.088$ |
| | 100 | $0.359 \pm 0.224$ | $0.641 \pm 0.198$ | $0.612 \pm 0.204$ | $0.686 \pm 0.131$ |
| Quadruped Run | 10 | $0.718 \pm 0.042$ | $0.727 \pm 0.023$ | $0.725 \pm 0.019$ | $0.731 \pm 0.013$ |
| | 30 | $0.642 \pm 0.107$ | $0.701 \pm 0.072$ | $0.702 \pm 0.074$ | $0.726 \pm 0.042$ |
| | 50 | $0.556 \pm 0.143$ | $0.679 \pm 0.135$ | $0.683 \pm 0.137$ | $0.705 \pm 0.078$ |
| | 100 | $0.398 \pm 0.217$ | $0.644 \pm 0.193$ | $0.629 \pm 0.184$ | $0.682 \pm 0.129$ |
| Quadruped Walk | 10 | $0.719 \pm 0.045$ | $0.728 \pm 0.027$ | $0.724 \pm 0.017$ | $0.732 \pm 0.011$ |
| | 30 | $0.656 \pm 0.092$ | $0.701 \pm 0.067$ | $0.703 \pm 0.072$ | $0.723 \pm 0.039$ |
| | 50 | $0.534 \pm 0.129$ | $0.682 \pm 0.114$ | $0.685 \pm 0.132$ | $0.701 \pm 0.082$ |
| | 100 | $0.413 \pm 0.212$ | $0.654 \pm 0.183$ | $0.634 \pm 0.179$ | $0.687 \pm 0.137$ |
| Reacher Easy | 10 | $0.711 \pm 0.065$ | $0.729 \pm 0.012$ | $0.724 \pm 0.013$ | $0.732 \pm 0.008$ |
| | 30 | $0.634 \pm 0.102$ | $0.706 \pm 0.061$ | $0.705 \pm 0.067$ | $\mathbf{0.731 \pm 0.031}$ |
| | 50 | $0.464 \pm 0.167$ | $0.675 \pm 0.104$ | $0.678 \pm 0.121$ | $0.720 \pm 0.075$ |
| | 100 | $0.373 \pm 0.221$ | $0.631 \pm 0.175$ | $0.614 \pm 0.163$ | $0.705 \pm 0.125$ |
| Reacher Hard | 10 | $0.712 \pm 0.072$ | $0.729 \pm 0.013$ | $0.724 \pm 0.014$ | $0.732 \pm 0.010$ |
| | 30 | $0.608 \pm 0.121$ | $0.702 \pm 0.062$ | $0.703 \pm 0.072$ | $\mathbf{0.730 \pm 0.042}$ |
| | 50 | $0.501 \pm 0.189$ | $0.663 \pm 0.114$ | $0.674 \pm 0.132$ | $0.717 \pm 0.082$ |
| | 100 | $0.349 \pm 0.233$ | $0.602 \pm 0.183$ | $0.607 \pm 0.172$ | $0.701 \pm 0.165$ |
| Walker Run | 10 | $0.702 \pm 0.137$ | $0.720 \pm 0.023$ | $0.713 \pm 0.027$ | $0.729 \pm 0.013$ |
| | 30 | $0.554 \pm 0.158$ | $0.676 \pm 0.067$ | $0.682 \pm 0.092$ | $0.711 \pm 0.072$ |
| | 50 | $0.424 \pm 0.213$ | $0.613 \pm 0.132$ | $0.658 \pm 0.188$ | $0.672 \pm 0.128$ |
| | 100 | $0.323 \pm 0.261$ | $0.532 \pm 0.191$ | $0.572 \pm 0.243$ | $0.645 \pm 0.175$ |
| Walker Stand | 10 | $0.706 \pm 0.087$ | $0.728 \pm 0.012$ | $0.721 \pm 0.012$ | $0.734 \pm 0.008$ |
| | 30 | $0.598 \pm 0.128$ | $0.690 \pm 0.051$ | $0.692 \pm 0.113$ | $0.722 \pm 0.067$ |
| | 50 | $0.494 \pm 0.173$ | $0.648 \pm 0.108$ | $0.658 \pm 0.162$ | $0.701 \pm 0.138$ |
| | 100 | $0.369 \pm 0.241$ | $0.568 \pm 0.172$ | $0.572 \pm 0.221$ | $0.662 \pm 0.185$ |
| Walker Walk | 10 | $0.701 \pm 0.118$ | $0.727 \pm \mathbf{0.015}$ | $0.723 \pm \mathbf{0.022}$ | $0.731 \pm 0.028$ |
| | 30 | $0.612 \pm 0.140$ | $0.696 \pm 0.063$ | $0.701 \pm 0.073$ | $\mathbf{0.730 \pm 0.034}$ |
| | 50 | $0.474 \pm 0.181$ | $0.664 \pm 0.123$ | $0.667 \pm 0.112$ | $0.718 \pm 0.079$ |
| | 100 | $0.371 \pm 0.251$ | $0.598 \pm 0.194$ | $0.601 \pm 0.171$ | $0.675 \pm 0.142$ |

## H.2 Trial Efficiency

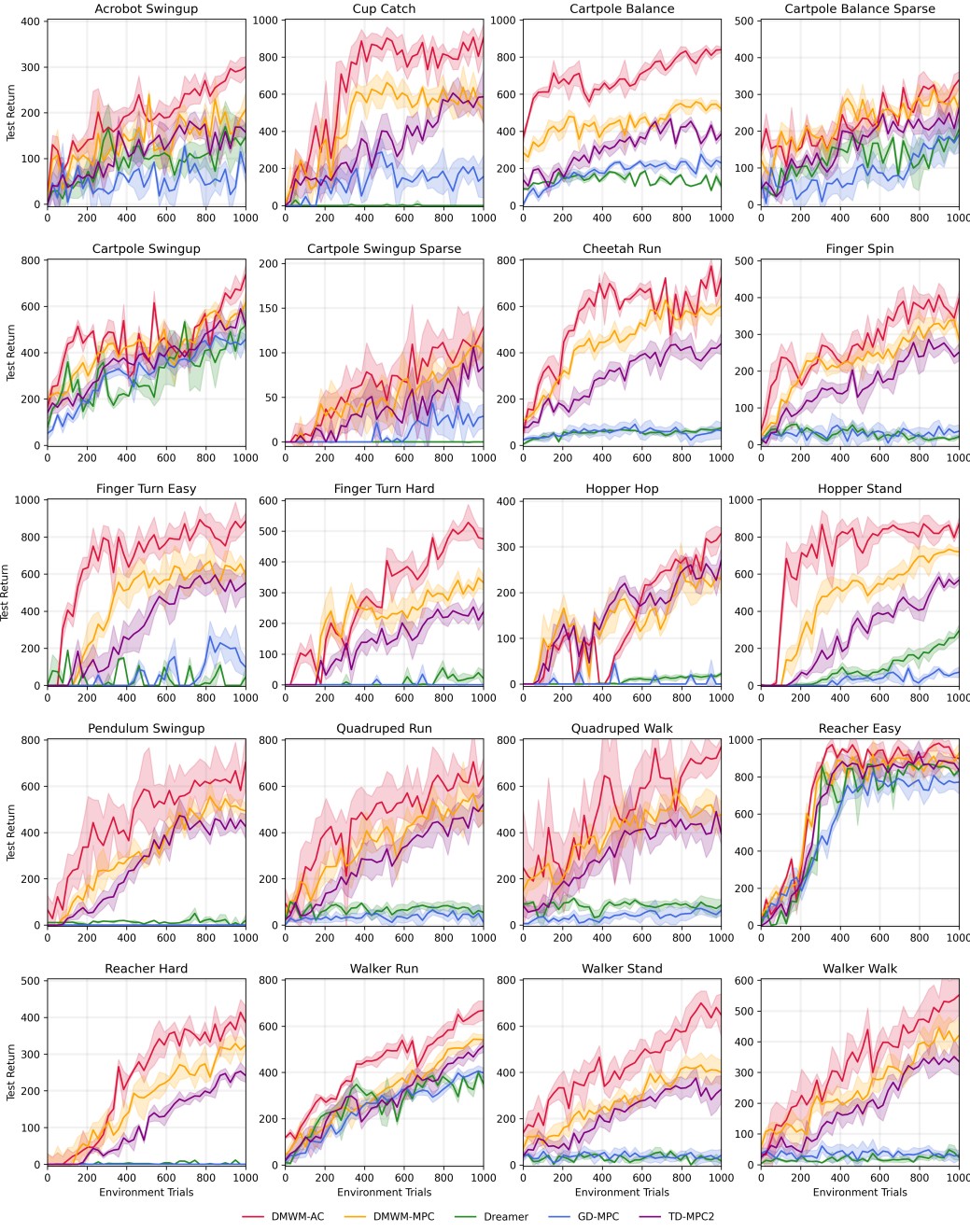

Figure 10: Performance comparison of test results on 20 DMC tasks under limited environment steps, where the standard error is shaded in the distraction setting. The horizontal axis indicates the number of environment data that is used to train the models. The vertical axis represents the average test return over 100 test episodes.

## H.3  Data Efficiency

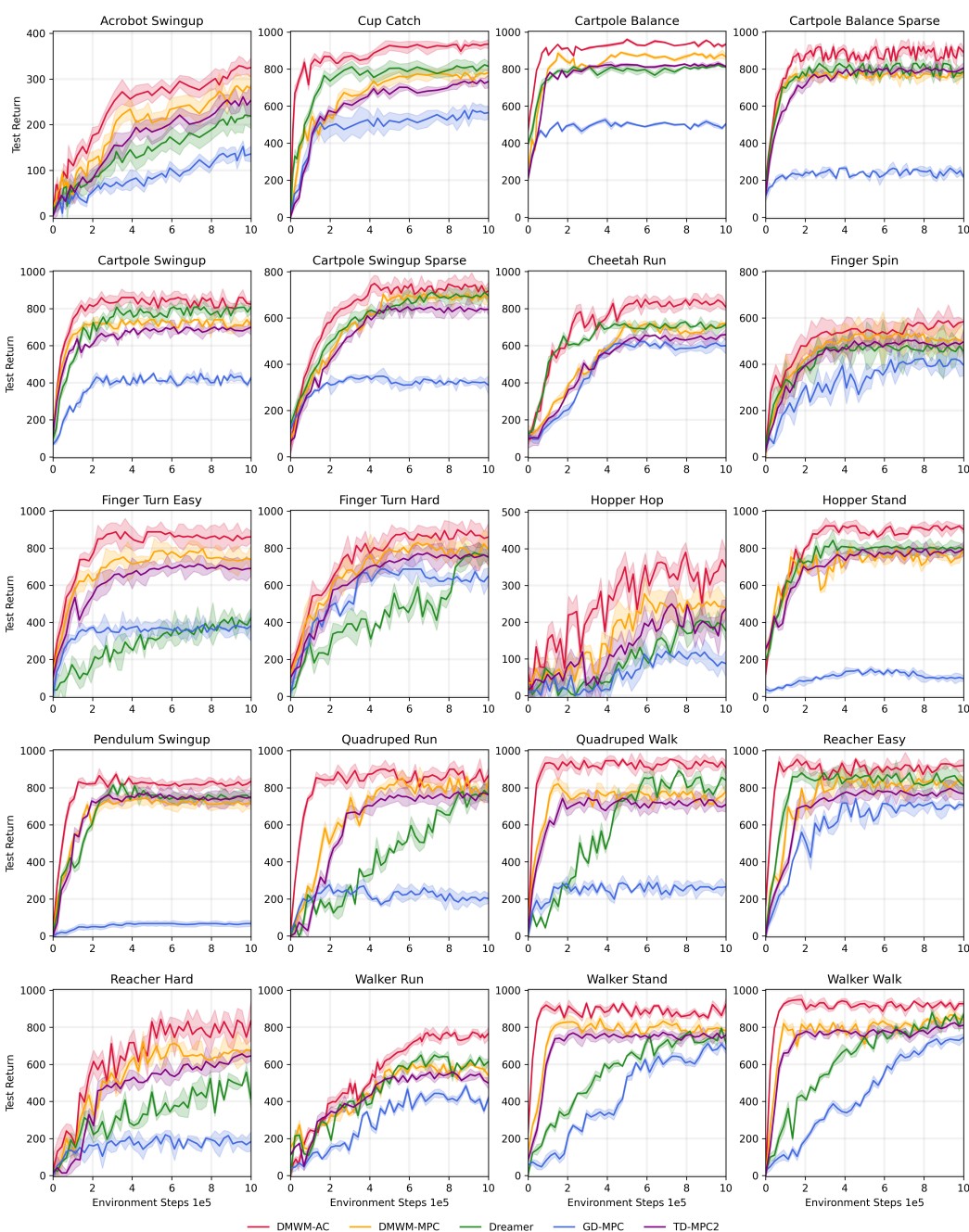

Figure 11: Performance comparison of test results on 20 DMC tasks under limited environment interactions, where the standard error is shaded in the distraction setting. The horizontal axis indicates the number of times that the models explore the environments. The vertical axis represents the average test return over 100 test episodes.

## H.4 Long-term Imaginations Over Extended Horizon Size

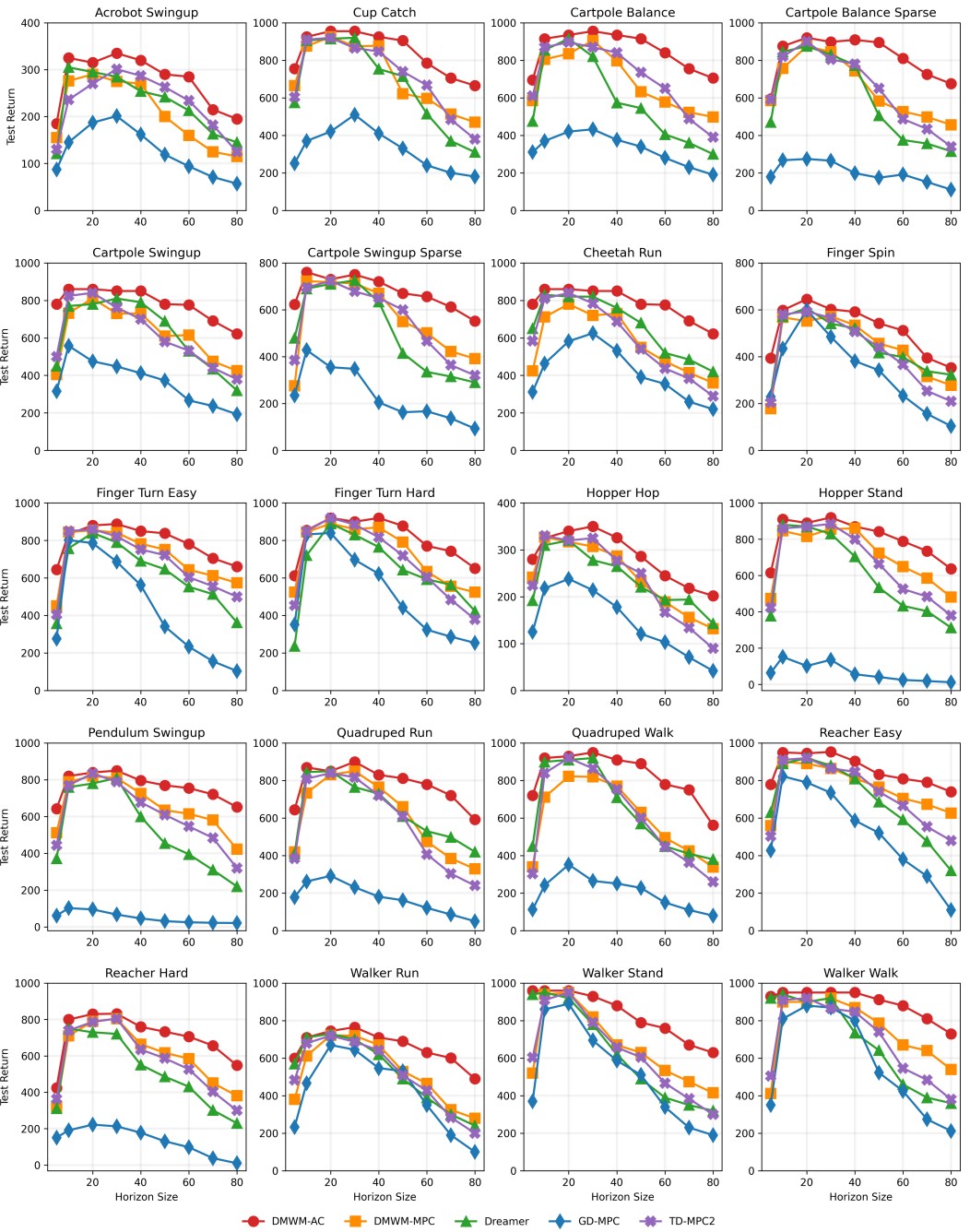

Figure 12: Performance comparison of test results on 20 DMC tasks over different horizon size of imagination. The horizontal axis indicates the horizon size of each imagination. The vertical axis represents the average test return over 100 test episodes.

## H.5 Impact of Logic Inference Depth Over Extended Horizon Size

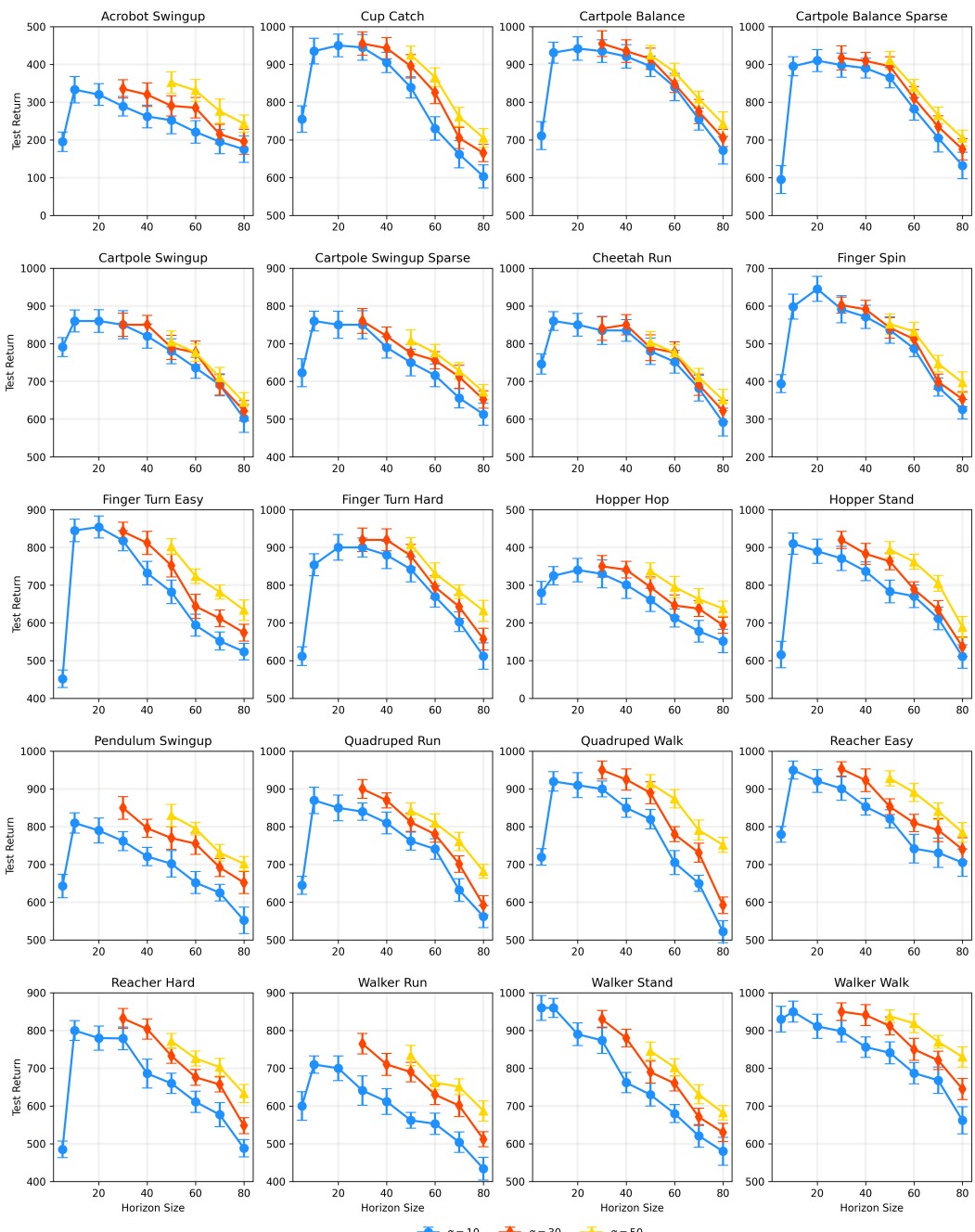

Figure 13: Performance comparison of test results on 20 DMC tasks with different logic inference depth $\alpha$ over extended horizon size of imagination. The horizontal axis indicates the horizon size of each imagination, and the vertical axis represents the average test return over 100 test episodes.

