# OpenReview forum: "DMWM: Dual-Mind World Model with Long-Term Imagination"
_NeurIPS.cc/2025/Conference — NeurIPS 2025 spotlight_

### Official Review · Reviewer_Y1Jx · 2025-06-19

**Clarity:** 3
**Significance:** 3
**Originality:** 3
**Rating:** 5
**Confidence:** 4

**Summary:**

The paper proposes a novel framework called Dual-Mind World Model (DMWM) to enhance the imagination capabilities of agents in complex environments. By integrating a recurrent state-space model (RSSM) with a logic-integrated neural network (LINN), DMWM aims to achieve logical consistency and long-term planning. The framework is evaluated on various benchmark tasks, demonstrating significant improvements in logical coherence, data efficiency, and trial efficiency compared to existing state-of-the-art models.

**Questions:**

1. How do the authors plan to extend the DMWM framework to environments with ambiguous or evolving logical rules?
2. What strategies will be adopted to mitigate the complexity introduced by the dual-system architecture in real-world applications?

**Ethical Concerns:**

["NO or VERY MINOR ethics concerns only"]

**Final Justification:**

As authors have addressed my concerns and proposed ideas about future work to address the limitations. I have raised my score correspondingly.

**Limitations:**

The paper acknowledges limitations regarding the model's reliance on predefined logical rules and its focus on specific tasks. Future work should aim to address these gaps and explore broader applications.

I would consider raising my score if the above concerns are properly addressed.

**Paper Formatting Concerns:**

This article has no formatting issues.

**Quality:**

3

**Strengths And Weaknesses:**

Strengths:
1. The dual-process framework effectively combines intuitive and logical reasoning, addressing limitations of traditional models that struggle with long-term predictions.
2. Comprehensive experiments on DMControl and robotic tasks showcase the framework's robustness and efficiency, providing a solid basis for its claims.
3. The ability to enhance exploration efficiency and performance in low-data regimes makes the proposed model relevant for real-world applications.

Weaknesses:
1. The reliance on predefined logical rules may limit adaptability to dynamic environments where such rules are not clear-cut. Although the authors have mentioned that in limitations, this remains a big concern about the generalizability and scalibility of the proposed dual-process methods.
2. Missing references. There are more relevant studies about dual-process models from human cognition and physical inference [1-4].
3. While the model performs well in controlled environments, its effectiveness in dynamic and unpredictable real-world scenarios remains uncertain.

[1] Li S, Wu K, Zhang C, et al. On the learning mechanisms in physical reasoning. NeurIPS 2022.

[2] Smith K, Battaglia P, Tenenbaum J. Integrating heuristic and simulation-based reasoning in intuitive physics. 2023.

[3] Sosa F A, Gershman S J, Ullman T D. Blending simulation and abstraction for physical reasoning. Cognition 2025.

[4] Li S, Ma Y, Yan J, et al. A Simulation-heuristics Dual-process Model for Intuitive Physics. CogSci 2025.

---

> ### Author Rebuttal · Authors · 2025-07-31
>
> We are grateful to Reviewer Y1Jx for spending time on reviewing our manuscript and raising many valuable improvement suggestions. In the response and our updated manuscript, we have made every effort to answer and address your insightful concerns. Our detailed responses are given as follows.
>
> > **W1:** The reliance on predefined logical rules may limit adaptability to dynamic environments where such rules are not clear-cut. Although the authors have mentioned that in limitations, this remains a big concern about the generalizability and scalability of the proposed dual-process methods.
>
> **A**:  We thank you for this comment that deserves further discussion. We first note that our method is general since the logical thinking ability of the System 2 component can be obtained by a learning-based method. In particular, the System 2 component be designed to learn logical rules from the actual environment changes with one-step logic $s_t \land a_t \rightarrow s_{t+1}$ and deep logical chain $s_{t-\alpha} \land a_{t-\alpha} \cdots \land s_t \land a_t \rightarrow s_{t+1}$. This learning-based method is applicable to any environment and any task without reliance on known and predefined logical rules. If there is an environment with clear rules, such as traffic rules for vehicles in autonomous driving, these rules can also scale to our proposed method as the logical regularization. Hence, our learning-based System 2 component is general and scalable to more complex applications. With regards to our claim regarding the reliance on domain-specific logic, this relates to the fact that action $a_t$ in $s_t \land a_t \rightarrow s_{t+1}$ follows the specific task $g$, i.e., $a \in \mathcal{A}_g \subsetneq \mathcal{A}$, and it leads to $s \in \mathcal{S}_g \subsetneq \mathcal{S}$, where $S$ is the state space of the open world, $\mathcal{A}$ is the action space of the agent, and $\mathcal{S}_g$ and $\mathcal{A}_g$ are respectively the subset of the state space and the action space with the task $g$. In this context, we can use the goal-conditioned method, similar to TD-MPC2 [1], to overcome this particular limitation and scale DMWM to different tasks with generalization.  In particular, DMWM will learn a task vector $g$ as an additional input to the System 1 component, such as $\hat{z}_t \sim p\left(\hat{z}_t \mid h_t, g\right)$. To demonstrate the generalization and scalability of the proposed method, we have added two experiments that extend DMWM to multi-task environments (scalability) and few-shot learning for new tasks (generalization). Due to the image and link limitations, we present data tables below for the experiment results. We first investigate the task performance in multi-task complex environments.
>
> | Method                | Env Steps | TD-MPC2     | DMWM (Proposed) |
> |----------------------|-------------|-------------|-----------------|
> | DMC Tasks (20 Tasks) | 1M          | 729 ± 11    | 785 ± 14        |
> |                      | 2M          | 787 ± 13    | 843 ± 12        |
> | ManiSkill2 (4 Tasks) | 1M          | 47 ± 3      | 61 ± 5          |
> |                      | 2M          | 56 ± 5      | 72 ± 5          |
> | MyoSuite (4 Tasks)   | 1M          | 53 ± 5      | 56 ± 3          |
> |                      | 2M          | 62 ± 6      | 68 ± 4          |
>
> In Table 1, the value for DMC tasks is episode return, and the value for ManiSkill2 and MyoSuite is success rate. Moreover, to support the multi-task environments, varying action dimensions across tasks by zero-padding are handled to a unified maximum dimension, and the action masking is applied to ignore invalid dimensions during training and inference [1]. The simulation results demonstrate that the proposed DMWM can be extended to multi-task environments based on the goal-conditioned method [1]. Compared to TD-MPC2, DMWM can achieve a performance improvement of 8% on DMC tasks, 30% on ManiSkill2, and 6% on MyoSuite at 1M environment steps by using the same hyperparameters across all tasks.
>
> Next, we investigate the generalization of the DMWM to new tasks, where the DMWM is trained on 16 DMC tasks and then finetuned to 4 new DMC tasks. The task vector $g$ is initialized by a semantically similar seen task.
>
> | Method           | *Env Steps* | From scratch | Finetuned      |
> |------------------|-------------|---------------|----------------|
> | Finger Turn Hard | 10k         | 11 ± 7        | 27 ± 14        |
> |                  | 20k         | 17 ± 9        | 43 ± 13        |
> | Quadruped Run    | 10k         | 9 ± 3         | 33 ± 12        |
> |                  | 20k         | 26 ± 15       | 51 ± 23        |
> | Reacher Hard     | 10k         | 13 ± 5        | 36 ± 11        |
> |                  | 20k         | 21 ± 8        | 41 ± 17        |
> | Hopper Hop       | 10k         | 3 ± 5         | 21 ± 19        |
> |                  | 20k         | 15 ± 6        | 27 ± 15        |
>
> The performance improvement across unseen tasks with few-shot learning demonstrates that DMWM can generalize effectively to novel environments. Compared to training from scratch, the finetuned model achieves higher scores under limited environment steps.
>
> References:
> [1] N. Hansen, H. Su, and X. Wang. Td-mpc2: Scalable, robust world models for continuous control. International Conference on Learning Representations, 2024.
>
> > **W2:** Missing references. There are more relevant studies about dual-process models from human cognition and physical inference [1-4].
>
> **A**: We have carefully read these references and have introduced them in our updated manuscript. These references are also valuable to our future work.
>
> > **W3:** While the model performs well in controlled environments, its effectiveness in dynamic and unpredictable real-world scenarios remains uncertain.
>
> **A**: While our experiments focus on controlled benchmarks such as the DMControl suite and robotic platforms, such as MyoSuite and ManiSkill2, we acknowledge that these environments, despite their complexity, may not fully capture the variability and stochasticity present in open-ended real-world scenarios. We have provided a more detailed discussion of this limitation and the potential solutions in the appendix (Please see the response to W2 of Reviewer Jdzu). In particular, for future work, to address the effectiveness of WMWM, we plan to develop the probabilistic logic to handle the ambiguous logical rules in real-world scenarios and develop the mechanisms for online logical rule adaptation to handle evolving and uncertain environments.
>
> > **Q1:** How do the authors plan to extend the DMWM framework to environments with ambiguous or evolving logical rules?
>
> **A**:  For ambiguous logical rules, we can further develop the probabilistic logic for the System 2 component. In particular, the existing logic similarity metric $\text{Sim}(\varphi, \text{True}) \in [0,1]$ for DMWM can be considered as a probabilistic estimate $\mathbb{P}(\varphi)$ of the truth value of a logical proposition $\varphi$. In this way, the probabilistic logic can be incorporated by introducing soft constraints of the form $\mathbb{P}(\varphi) \geq \tau$, where $\tau$ is a required confidence threshold. These constraints can be integrated into the logic loss through regularization terms or logic-enhanced variational objectives that reflect the probabilistic consistency of imagined trajectories. To address the evolving logical rules, DMWM can incorporate meta-learning or continual-learning mechanisms to dynamically update logic rules.
>
> > **Q2:** What strategies will be adopted to mitigate the complexity introduced by the dual-system architecture in real-world applications?
>
> **A**: One of the most promising strategies can be selective logical Inference through uncertainty estimation. In particular, to reduce the computational overhead of always-on logical reasoning, we can use uncertainty-aware control mechanisms, such as the mentioned probabilistic logic in the response to your Q1 comment, to selectively activate the System 2 component. In particular, when the System 1 component exhibits high epistemic or aleatoric uncertainty in prediction (e.g., via ensemble variance or Monte Carlo dropout), logical reasoning from the System 2 component will be activated to guide or correct the imagined trajectory. This adaptive triggering mimics the cognitive allocation of deliberative reasoning in humans and significantly reduces redundant logical inference in seen situations. Moreover, we can adopt sparse and pruned logic representations. In particular, as the logic heatmap in Figure 3, we have found that many logical rules can be sparsely activated since the logical operations within the System 2 component are executed over a structured set of vectorized embeddings. This enables effective pruning or attention-based selection of relevant rules during inference, which can provide a reduced computation graph without losing consistency.

---

> > ### Comment · Reviewer_Y1Jx · 2025-08-03
> >
> > Thank you for addressing my concerns. Overall, I think it's an interesting and novel work.

---

### Official Review · Reviewer_2YL1 · 2025-07-01

**Clarity:** 4
**Significance:** 3
**Originality:** 3
**Rating:** 5
**Confidence:** 4

**Summary:**

This paper presents a neurosymbolic model of dual-process theory in human cognition, introducing DMWM, which combines an RSSM-based System 1 for fast, intuitive state prediction with a logic-integrated neural network (LINN)-based System 2 for higher-level logical reasoning. LINN-S2 imposes logical constraints (via regularization rules) on the state and action spaces to ensure that RSSM-S1's imagined trajectories comply with domain-specific logic. The model is evaluated in both actor-critic reinforcement learning and model predictive control settings.

**Questions:**

Q1: Regarding the use of the Kronecker product in LINN-S2, this work applies Kronecker product to combine the latent state and action vectors before logic processing. What motivated this choice over more conventional fusion strategies like concatenation or bilinear pooling? Did you experiment with alternatives, and how does the Kronecker product impact downstream logical reasoning or model stability?

Q2: Did you explore how DMWM performs when some logic rules are systematically incorrect, redundant, or even contradictory?

Q3: Why apply logical regularization on joint embeddings rather than over predicted transitions? Could you elaborate on why you chose to apply logic at this representational level, and whether other points of intervention (e.g., such as over $(s_t, s_{t+1})$ pairs or full imagined trajectories) were considered or tested?

**Ethical Concerns:**

["NO or VERY MINOR ethics concerns only"]

**Final Justification:**

My concerns have been cleared with the additional experiments.

System 1/2 is an overloaded term. This work sets a much better standard of what the machine learning community would expect in a technical paper. Unlike many symbolic-augmented systems where logic is only used post hoc, DMWM features mutual feedback. System 1 provides latent rollouts that inform rule discovery, while System 2 enforces logical constraints that correct and regularize System 1's behavior. This interleaving helps maintain coherence, especially as authors argue about long-term imagination. I recommend acceptence of this work.

**Limitations:**

Yes

**Quality:**

3

**Strengths And Weaknesses:**

## Strengths

* **Mutual feedback**: Unlike many symbolic-augmented systems where logic is only used post hoc, DMWM features mutual feedback. System 1 provides latent rollouts that inform rule discovery, while System 2 enforces logical constraints that correct and regularize System 1's behavior. This interleaving helps maintain coherence, especially as authors argue about long-term imagination. This is very good design philosophy.
* **Experiments**: The experiments show that DMWM improves consistency and robustness, suggesting the logic-regularized component improves extrapolation under uncertainty.

## Weaknesses

* Heavy reliance on manually defined logic rules. The logic component depends on domain-specific rule definitions, which may not generalize across environments. While authors briefly mentioned this and gave a discussion on the limitations, this paper can benefit from more analysis on how the system would perform with imperfect, noisy, or missing rules, and no ablations to test sensitivity to these conditions. Solutions (e.g., fuzzy logic?) are not expected, but it would be a plus to see the failure patterns of this method.
* The dual-module design, including Kronecker-product feature interactions and recursive logical rollouts, adds significant architectural and computational complexity. The paper does not explore how well this framework scales to high-dimensional or more open-ended tasks where symbolic abstraction may be harder to define. It can be hard to scale this work up to a modern scale.
* Authors should provide more quantitative and qualitative comparisons to the line of work mentioned in Appendix F2 (complex task planning [71, 72, 25, 73, 26, 74, 75]), including Logic Tensor Networks. Also LS‑Imagine (Li et al., ICLR 2025) and other jumpy state-space methods offer similar long-range planning benefits.

- - -

Open-World Reinforcement Learning over Long Short-Term Imagination. Jiajian Li, Qi Wang, Yunbo Wang, Xin Jin, Yang Li, Wenjun Zeng, Xiaokang Yang. ICLR 2025 Oral.

---

> ### Author Rebuttal · Authors · 2025-07-31
>
> We are grateful to Reviewer 2YL1 for spending time on reviewing our manuscript and raising many valuable improvement suggestions. Our detailed responses are given as follows.
>
> > **W1:** Heavy reliance on manually defined logic rules.
>
> **A**: Thank you for this feedback. First of all, our method is general since the logical thinking ability of the System 2 component can be obtained by a learning-based method instead of hand-crafted rules. In particular, the System 2 component learns logics from the actual environment changes with one-step logic $s_t \land a_t \rightarrow s_{t+1}$ and multi-step logical chain $s_{t-\alpha} \land a_{t-\alpha} \cdots \land s_t \land a_t \rightarrow s_{t+1}$. This learning-based method is applicable to any environment and any task without reliance on known and predefined logical rules. To demonstrate the generalization and scalability of the proposed method, we have added two experiments that extend DMWM to multi-task environments (scalability) and few-shot learning for new tasks (generalization) (Please see our response to W1 of Reviewer Y1Jx). Moreover, we provide more analysis of when DMWM performs well/not well (Please see our response to W2 of Reviewer Jdzu).
>
> > **W2:** The dual-module design, including Kronecker-product feature interactions and recursive logical rollouts, adds significant architectural and computational complexity. The paper does not explore how well this framework scales to high-dimensional or more open-ended tasks where symbolic abstraction may be harder to define. It can be hard to scale this work up to a modern scale.
>
> **A**: While the integration of Kronecker product-based feature interactions and recursive logical reasoning adds architectural complexity, these operations are confined to compact logical embeddings rather than raw high-dimensional spaces. Moreover, System 2 operates at an abstract symbolic level that scales with task logic rather than input dimensionality. Hence, the architectural and computational complexity of DMWM is controllable and moderate. Moreover, as mentioned in response to W1, the logical rules learned by the System 2 component are general and can be extended to an open-ended environment while keeping the scalability for specific rules.
>
> > **W3:** Authors should provide more quantitative and qualitative comparisons to the line of work mentioned in Appendix F2 (complex task planning [71, 72, 25, 73, 26, 74, 75]), including Logic Tensor Networks. Also, LS‑Imagine (Li et al., ICLR 2025) and other jumpy state-space methods offer similar long-range planning benefits.
>
> **A**: We are grateful for this feedback. In the revised manuscript, we have first added more detailed introductions of S‑Imagine (Li et al., ICLR 2025) and other jumpy state-space methods as the important complex planning methods in Appendix F2. Regarding to your concerns about quantitative and qualitative comparisons with [71, 72, 25, 73, 26, 74, 75], one of our main contributions is to propose a general dual-mind world model, in which we can replace LINN with any logical method for the System 2 component. Hence, the choice of the best logical inference method is not our research focus. The jumpy state-space methods are important to realize the long-term planning, especially the long short-term imagination proposed in (Li et al., ICLR 2025). For control tasks that require accurate single-step actions,  the quantitative comparison of DMWM and LS-imagine via environment trails in 4 representative DMC tasks is shown as follows, where environment trials represent the number of environment exploration opportunities.
>
> | Method           | Env Trials | DMWM (Proposed) | LS‑Imagine       |
> |------------------|--------------|------------------|------------------|
> | Cartpole Balance | 200     | 574 ± 24 | 321 ± 32   |
> |  | 400 | 643 ± 26 | 423 ± 27 |
> |  | 600 | 721 ± 18 | 532 ± 28 |
> |  | 800 | 787 ± 29  | 654 ± 23 |
> |  | 1000 | 837 ± 17 | 764 ± 19 |
> | Finger Turn Easy | 200 | 531 ± 22 | 192 ± 17 |
> |  | 400 | 712 ± 19 | 571 ± 23 |
> |  | 600 | 749 ± 14   | 621 ± 18 |
> |  | 800 | 789 ± 19  | 678 ± 13 |
> |  | 1000 | 765 ± 21   | 702 ± 28  |
> | Hopper Stand | 200 | 568 ± 29 | 178 ± 17 |
> | | 400 | 721 ± 21 | 506 ± 25 |
> | | 600 | 802 ± 25 | 623 ± 19 |
> | | 800 | 812 ± 14 | 713 ± 28 |
> | | 1000 | 821 ± 16 | 732 ± 28 |
> | Walker Stand | 200  | 278 ± 31 | 98 ± 28 |
> | | 400  | 409 ± 33  | 226 ± 35 |
> | | 600 | 482 ± 23| 310 ± 27 |
> | | 800 | 598 ± 31| 387 ± 40 |
> | | 1000 | 685 ± 33| 517 ± 28 |
>
> The simulation results demonstrate that the proposed DMWM has better exploration efficiency compared to LS-imagine since LS-image is still a pattern-driven Intuitive approach, i.e., System 1 defined in our paper. Next, the qualitative comparison is given as follows.  DMWM and LS‑Imagine both aim to improve long-horizon imagination, with different paradigms. LS‑Imagine relies on jumpy transitions and affordance maps to extend the imagination horizon in visually grounded environments. It enables efficient exploration through value-guided behaviors for an open world. In contrast, DMWM introduces a dual-mind architecture inspired by human cognition, which combines intuitive RSSM-based modeling with a logic-integrated neural network that enforces symbolic reasoning and consistency over an extended horizon. While LS‑Imagine performs spatially-aware exploration using visual semantics, DMWM provides interpretable and semantically coherent predictions through structured logical inference.
>
> > **Q1:** Regarding the use of the Kronecker product in LINN-S2, this work applies Kronecker product to combine the latent state and action vectors before logic processing. What motivated this choice over more conventional fusion strategies like concatenation or bilinear pooling? Did you experiment with alternatives, and how does the Kronecker product impact downstream logical reasoning or model stability?
>
> **A**: We agree that the motivation needs a more detailed discussion. We must first point out that we had a typo in the manuscript. In particular, $\otimes$ represents outer product instead of Kronecker product since the state embedding $v \in \mathbb{R}^d$ and the action embedding $m \in \mathbb{R}^d$ are vectors. In other words, for vectors, $\text{OP}(v,m) = \text{KP}(v,m).\text{reshape}(d,d) \in \mathbb{R}^{d \times d}$, where $\text{OP}$ is the outer product operator and $\text{KP}$ is the outer product operator. In this context, our method of the LINN-S2 component that uses outer product and the convolutional neural network (CNN)  to capture the cross-space logical relationships is similar to bilinear pooling [1], which also captures multiplicative interactions between feature pairs. Moreover, in the appendix, to verify the stability and logical consistency of the outer product, we have added an ablation study to demonstrate the superior performance of the outer product compared to concatenation  (Please see our response to W1 of Reviewer YoYe). The simulation results demonstrate that the outer product of DMWM can improve logical consistency compared to the concatenation method across all tasks. This is because, compared to the concatenation method that overlooks structural dependencies, the outer product captures second-order interactions between latent state and action embeddings for cross-space logic modeling. Hence, the cross-space representation by outer product and CNN can enable the System 2 component to express more complex logical dependencies essential for long-horizon imagination. As demonstrated in the experiment results, this design is effective and stable for logic processing in our framework.
>
> References:
> [1] Kong S, et al. Low-rank bilinear pooling for fine-grained classification. Proc IEEE CVPR, 2017: 365–374.
>
> > **Q2:** Did you explore how DMWM performs when some logic rules are systematically incorrect, redundant, or even contradictory?
>
> **A**: Yes. As our response to W1, the System 2 component is a learning-based method to capture logic rules from the environment changes. Hence, if there are systematically incorrect, redundant, or contradictory logic rules, it means that the observations from the environment are noisy or extremely stochastic. In this context, we can incorporate meta-learning or continual-learning mechanisms to dynamically update logic rules or introduce probabilistic logic to reflect the probabilistic consistency of trajectories. We have provided more details and potential solutions for this discussion in the appendix (Please see the response to W2 of Reviewer Jdzu).
>
> > **Q3:** Why apply logical regularization on joint embeddings rather than over predicted transitions? Could you elaborate on why you chose to apply logic at this representational level, and whether other points of intervention (e.g., such as over pairs $\boldsymbol{(s_t, s_{t+1})}$ or full imagined trajectories) were considered or tested?
>
> **A**:  To clarify, the logic learning can be divided into two parts: one is logical rules from actual environment trajectories, and the other is logical regularization from basic logical laws. In particular, the purpose of logical regularization is to ensure that the neural modules implement logical operations (i.e., $\text{AND}$, $\text{OR}$, and $\text{NOT}$) following basic logical laws such as *identity*, *annihilator* and *idempotence*, as shown in Appendix G. The logical regularizers are applied over individual vectors $w$ rather than over joint embeddings. Theoretically, $w$ can be any vector, since logical regularization only aims to ensure that the logical networks behave consistently under logical transformations, regardless of the specific origin of input vectors. Hence, logical operations require semantically structured inputs, while the raw state lacks learned logic structures. Moreover, embedding both actions and states into a shared logic space enables compositional reasoning across spaces, which can be explicitly optimized for symbolic logical rules.

---

> > ### Comment · Reviewer_2YL1 · 2025-07-31
> >
> > Thanks for your response. My concerns have been cleared. I would like to maintain my evaluation (5: Accept) and recommend this good paper to appear in NeurIPS.

---

### Official Review · Reviewer_Jdzu · 2025-07-02

**Clarity:** 3
**Significance:** 4
**Originality:** 3
**Rating:** 5
**Confidence:** 3

**Summary:**

Reinforcement learning algorithms have become increasingly important in the last years. One of the major challenges to deep reinforcement learning algorithms is sample efficiency, especially for long-term tasks. To address this issue many approaches learn a world model to train an agent entirely in imagination, eliminating the need for direct environment interaction during training. However, these methods often suffer from either a lack of imagination accuracy (accumulation of prediction errors), exploration capabilities, or runtime efficiency.

The paper proposes DMWM, a model framework that seeks to integrate logical reasoning to enable imagination with logical consistency to improve training in imagination. The model consists of (i) a component (RSSM-S1 based on DreamerV3) that handles state transitions and (ii) a logic-integrated neural network-based System (LINN-S2) that guides the imagination process by including logical reasoning.

Numerical experiments are performed for the DeepMind control suite as well as 4 robotic tasks from ManiSkill2 platform and 4 robotic tasks from MyoSuite platform. Results are compared to SOTA baseline solutions (Dreamer, Hieros, HRSSM).

**Questions:**

1. What are the main insights? Under which conditions (e.g. size/dimensions of the state space or action space) the framework performs good / not good? Where does the good performance come from? Which components are responsible?
2. Does the proposed technique scale to larger problems (dimensions of the state space or action space)? How many training will be required? Please discuss.
3. What are the required runtimes for the performed experiments?
4. How problematic is the tuning of the hyperparameters used?
5. Could the operations used to describe logical rules be extended?

**Ethical Concerns:**

["NO or VERY MINOR ethics concerns only"]

**Final Justification:**

The authors addressed all questions I had but given I already rated this paper "Accept", I will not change my rating. It's not exceptional but incremental (in a good way).

**Limitations:**

yes

**Paper Formatting Concerns:**

I understand that there is a lack of space due to the page limit. However, the paper is squeezed at various places.

* Line 73: The logical rules allows  $\to$ allow
* Line 619: TABLE 2 $\to$ Table 3
* Line 621: TABLE 3 $\to$ Table 4
* Line 693: TABLE 5 $\to$ Table 5
* 696: “Pendulum Swingup“ too wide (page 23)

**Quality:**

3

**Strengths And Weaknesses:**

Strengths:
 * The proposed model is novel and creative. The investigated problem is timely and highly relevant.
 * The provided benchmark experiments against existing state-of-the-art models are rich and a strong performance is demonstrated.
 * The paper is well written and well executed.


Weaknesses:
 * The logical rules have to be known and predefined, which is fine for some applications but a limiting factor for others.
 * Limitations of the approach could be better discussed. When does the framework perform particularly good / not good?

---

> ### Author Rebuttal · Authors · 2025-07-31
>
> > **W1:** The logical rules have to be known and predefined, which is fine for some applications but a limiting factor for others.
>
> **A**: Thank you for this comment. Our proposed method is general and scalable since the logical thinking ability of the System 2 component can be obtained by a learning-based method instead of hand-crafted rules. In particular, the System 2 component is designed to learn logical rules from the actual environment trajectories $s_t \land a_t \rightarrow s_{t+1}$. This method is applicable to any environment and any task without reliance on prior knowledge of logic. To demonstrate the generalization and scalability of the proposed method, we have added two experiments that extend DMWM to multi-task environments and few-shot learning for new tasks (Please see our response to W1 of Reviewer Y1Jx).
>
> > **W2:** Limitations of the approach could be better discussed. When does the framework perform particularly good/ not good?
>
> **A**: We first detail the cases in which DMWM performs well:
> - Environments with Clear Structural Rules: DMWM uses logic rules most effectively in domains where domain knowledge (e.g., physical constraints or control dependencies) can be formalized through first-order logic. For instance, in the *Cartpole Balance* task from the DMControl suite, domain rules like “if the pole angle exceeds a threshold, the agent must apply a force in the opposite direction to maintain balance” can be encoded as logical implications, i.e., `angle > θ → force = left`. In such environments with clear structural rules, logical regularization and learned logical rules can improve both interpretability and generalization.
> - Complex Tasks Requiring Long-Horizon Planning: RSSM-S1 performs well in short-term prediction but cannot handle the error accumulation over longer horizons. LINN-S2 enhances recursive implication reasoning to construct long-term logical chains and aligns the predicted trajectories with logical inference, as shown in (10) and (11). This reliable imagination ability is important for complex tasks with the requirements of long-horizon planning. For instance, in tasks such as *Quadruped Run* in Figure 13 and *Key Turn Hard* in Figure 12, DMWM achieves more stable long-horizon planning compared to Dreamer.
> - Limited-Data or Limited-Trial Settings: Unlike conventional RSSM-based models that rely solely on statistical patterns in observed transitions, LINN-S2 imposes logical constraints that function as supervisory signals. These logic-driven regularizations make the model generalize effectively from sparse or partial observations, which enables more efficient learning of long-horizon dynamics. As shown in Figures 9–12, DMWM outperforms baselines with substantially fewer environment interactions.
>
> Next, we discuss some of the scenarios in which DMWM may have limited performance gains:
> - Simple Tasks with Short-Term Dependencies: In simple tasks that are characterized by short-term dependencies and low-dimensional dynamics, such as *Cartpole Balance* or *Cup Catch*, DMWM takes limited advantage over baseline models, as shown in Figure 10. The additional logical reasoning from LINN-S2 does not significantly enhance prediction quality, as the short-term dynamics can be effectively captured by RSSM-S1 alone. In such cases, the complexity and computational overhead introduced by hierarchical logical inference are over the task demands.
> - Highly Stochastic Environments: In highly stochastic environments, LINN-S2 faces difficulty in establishing effective logical dependencies between states and actions. Although LINN-S2 learn logical rules from general environment transitions $s_t \land a_t \rightarrow s_{t+1}$, this still relies on the presence of stable, interpretable relationships, which are often absent in high-stochastic settings with sparse logic rules. In this context, the logic consistency loss $C(\phi)$ can be noisy and unreliable. To address this, future work can incorporate uncertainty-aware logic modeling or probabilistic logic reasoning to selectively apply logical constraints only where they are supported by sufficient structural confidence.
> - Non-Stationary Environments: In non-stationary environments where underlying logical dependencies evolve over time, the static formulation of LINN-S2 fails to maintain accurate inference since it cannot adapt to abrupt changes in environment dynamics. This failure can lead to outdated or incorrect logical constraints being imposed on RSSM-S1 and lead to degraded imagination quality, along with unstable planning. Future extensions can incorporate continual learning to dynamically update logic rules.
>
> We will add this discussion to the appendix.
>
> > **Q1** What are the main insights? Under which conditions (e.g. size/dimensions of the state space or action space) the framework performs good / not good? Where does the good performance come from? Which components are responsible?
>
> **A**: We agree that this issue deserves more discussions, as explained next. Our work proposes a dual cognitive system with fast, data-driven System 1 and logic-driven System 2, which enables robust long-horizon imagination by mitigating the accumulated prediction errors. This is achieved through a recursive logic reasoning framework that imposes local and global logical consistency on temporal predictions to improve long-term planning. The inter-system feedback mechanism ensures that symbolic logical rules can refine intuitive transitions and adapt logic based on environmental feedback. The conditions of performance can be seen in our response to W2. Performance gains primarily arise from the LINN-S2 component, which learns logical rules from the environment through hierarchical deep reasoning, thereby ensuring logical consistency in imagined rollouts. The inter-system feedback mechanism enables the intuitive predictions of RSSM-S1 to align with logical constraints, thus creating a closed-loop dual-mind learning process. Overall, the dual-mind design and the inter-system feedback are responsible for performance gains.
>
> > **Q2:** Does the proposed technique scale to larger problems (dimensions of the state space or action space)? How many training will be required? Please discuss.
>
> **A**: Yes, DMWM can scale to larger problems due to its modular dual-system architecture. In particular, RSSM-S1 captures environment dynamics through compact latent representations to generalize across high-dimensional observations. Meanwhile, the LINN-S2 enhances scalability by reasoning over logic-structured embeddings rather than raw state-action inputs, which reduces the dimensional complexity of inference. The use of logic operations in LINN-S2 abstracts high-dimensional dependencies into compact, symbolic representations. It maintains reasoning tractability even as the state/action dimension increases. Moreover, the proposed hierarchical reasoning mechanism (10) enables the model to recursively infer over long horizons without unrolling full trajectories, which also alleviates the computational overhead in large-scale environments.
>
> > **Q3:** What are the required runtimes for the performed experiments?
>
> **A**: All experiments are conducted on a single NVIDIA RTX 3090 GPU. Training a single DMC task with DMWM requires 1.2 GPU days, compared to 0.35 GPU days for a standard RSSM. It reflects the additional cost of dual-system optimization and inter-system feedback. However, DMWM introduces no additional computational overhead for inference after training since S1 and S2 function as a unified model, with logical reasoning distilled into RSSM-S1. Hence, DMWM maintains runtime complexity comparable to RSSM.
>
> > **Q4:** How problematic is the tuning of the hyperparameters used?
>
> **A**: The tuning of hyperparameters is moderately complex. In particular, hyperparameters stem from RSSM-S1,  LINN-S2, and inter-system feedback. While RSSM-S1 builds on existing hyperparameters from architectures like DreamerV3, LINN-S2 introduces additional tuning needs, such as reasoning depth and logic regularization weights. However, these are guided by logical rules, which reduce arbitrariness and improve stability. The inter-system feedback mechanism requires balancing, but is supported by clearly defined loss components. Overall, although tuning DMWM involves more components than standard world models, the process remains feasible and justified.
>
> > **Q5:** Could the operations used to describe logical rules be extended?
>
> **A**: Yes, they could. As we have implemented the propositional logic system with $\neg$, $\land$, $\lor$, and $\to$ in a modular way, it is easy to form more complex, compositional logical operations within propositional logic.  Moreover, the operations can also be extended to first-order logic ($\forall$ and $\exists$) and probabilistic logic ($P(\varphi)$). In particular, the universal quantification $\forall$ can be extended by $\forall x: \varphi(x) \to \min_{x \in D} \text{Sim}(\varphi(x), \text{True})$, and the existential quantification $\forall$ can be extended by   $\exists x: \varphi(x) \Rightarrow \max_{x \in D} \text{Sim}(\varphi(x), \text{True})$, where $D$ represents the domain of variable $x$ and $\varphi(x)$ represents a logical formula evaluated over $x$. The min-pool and max-pool can be added to the logic loss to train the $\forall$ and $\exists$, respectively. To extend operations to probabilistic logic, the existing logic similarity metric $\text{Sim}(\varphi, \text{True}) \in [0,1]$ for DMWM can be considered as a probabilistic estimate $P(\varphi)$ of the truth value of a logical proposition $\varphi$. In this way, the probabilistic logic can be incorporated by introducing soft constraints of the form $P(\varphi) \geq \tau$, where $\tau$ is a required confidence threshold. These constraints can be integrated into the logic loss through variational objectives that reflect the probabilistic consistency of imagination.

---

### Official Review · Reviewer_YoYe · 2025-07-05

**Clarity:** 3
**Significance:** 3
**Originality:** 3
**Rating:** 5
**Confidence:** 3

**Summary:**

This work proposes a neural-symbolic world model consists of system I and II for long-term imagination and planning. The system I is built upon on  RSSM from DreamerV3. The system II is based on Logic-integrated neural network that implement hand-crafted propositional logical rules using neural networks, incorporating logical consistency across imagination states. System I and system II are trained jointly via inter-system feedback: system II is trained with system I's predicted latent dynamics; system I minimizes observation loss and the logical constraint loss from system II. The authors conduct experiments on robotics and control tasks. The results show that the proposed method outperform existing world models in terms of test return and logical consistency.

**Questions:**

How did you measure the logical consistency for evaluation? did you use the trained system II?

**Ethical Concerns:**

["NO or VERY MINOR ethics concerns only"]

**Final Justification:**

I'm more convinced that this paper should be accepted.

**Limitations:**

yes

**Quality:**

3

**Strengths And Weaknesses:**

## Strengths

* This work is well motivated: incorporating logical constraints into world model is an important research problem.

* It presents an interesting and novel approach to modeling world dynamics through neural-symbolic learning.

* The experiment results show the effectiveness of the proposed method.

## Weaknesses

* As discussed in the limitation section, the method relies on hand-crafted rules, which are difficult to scale or fully cover in more complex environments.

* The logical rules considered are restricted to propositional logic. It's nontrivial to extend this approach to more expressive logic with existential quantifiers and uncertainty.

* (minor) The authors claim that their application of Kronecker product can facilitate action-state alignment, but there lacks an ablation study to support this claim.

---

> ### Author Rebuttal · Authors · 2025-07-31
>
> We are grateful to Reviewer YoYe for spending time on reviewing our manuscript and raising many valuable improvement suggestions. In the response and our updated manuscript, we have made every effort to answer and address your insightful concerns. Our detailed responses are given as follows.
>
> > **W1:** As discussed in the limitation section, the method relies on hand-crafted rules, which are difficult to scale or fully cover in more complex environments.
>
> **A**: Thank you for this feedback. Our proposed method is general and scalable since the logical thinking ability of the System 2 component can be obtained by a learning-based method instead of hand-crafted rules. In particular, the System 2 component is designed to learn logical rules from the actual environment trajectories $s_t \land a_t \rightarrow s_{t+1}$. This method is applicable to any environment and any task without reliance on prior knowledge of logic. With regards to our claim regarding the reliance on domain-specific logic, this relates to the fact that action $a_t$ in $s_t \land a_t \rightarrow s_{t+1}$ follows the specific task $g$, i.e., $a \in \mathcal{A}_g \subsetneq \mathcal{A}$, and it leads to $s \in \mathcal{S}_g \subsetneq \mathcal{S}$, where $S$ is the state space of the open world, $\mathcal{A}$ is the action space of the agent, and $\mathcal{S}_g$ and $\mathcal{A}_g$ are respectively the subset of the state space and the action space with the task $g$. In this context, we can use a goal-conditioned method, similar to TD-MPC2 [1], to overcome this particular limitation and scale to different tasks. To demonstrate the generalization and scalability of the proposed method, we have added two experiments that extend DMWM to multi-task environments (scalability) and few-shot learning for new tasks (generalization) (Please see our response to W1 of Reviewer Y1Jx). The simulation results demonstrate that DMWM can achieve a performance improvement of 8% on DMC tasks, 30% on ManiSkill2, and 6% on MyoSuite compared to TD-MPC2 at 1M environment steps by using the same hyperparameters across all tasks. Moreover, we have added one paragraph that discusses, in detail, the limitations of our method in the appendix for clarifying these points. In a nutshell, our approach can be general and scalable for complex environments.
>
> References:
>
> [1] N. Hansen, H. Su, and X. Wang. Td-mpc2: Scalable, robust world models for continuous control. International Conference on Learning Representations, 2024.
>
> > **W2:** The logical rules considered are restricted to propositional logic. It's nontrivial to extend this approach to more expressive logic with existential quantifiers and uncertainty.
>
> **A**: We have adopted the propositional logic system since it is enough to capture the general environment logical rules by $s_t \land a_t \rightarrow s_{t+1}$. However, DMWM  can also be extended to existential quantifiers ($\forall$ and $\exists$) and probabilistic logic ($\mathbb{P}(\varphi)$). In particular, the universal quantification $\forall$ can be extended by   $\forall x: \varphi(x) \Rightarrow \min_{x \in D} \text{Sim}(\varphi(x), \text{True})$, and the existential quantification $\forall$ can be extended by   $\exists x: \varphi(x) \Rightarrow \max_{x \in D} \text{Sim}(\varphi(x), \text{True})$, where $D$ represents the domain of variable $x$ and $\varphi(x)$ represents a logical formula evaluated over $x$. The min-pool and max-pool can be added to the logic loss to train $\forall$ and $\exists$, respectively. To enable DMWM with logic uncertainty, the existing logic similarity metric $\text{Sim}(\varphi, \text{True}) \in [0,1]$ in (12) can be considered as a probabilistic estimate $\mathbb{P}(\varphi)$ of the truth value of a logical proposition $\varphi$. In this way, probabilistic logic can be incorporated by introducing soft constraints of the form $\mathbb{P}(\varphi) \geq \tau$, where $\tau$ is a required confidence threshold. These constraints can be integrated into the logic loss through regularization terms or logic-enhanced variational objectives that reflect the probabilistic consistency of imagined trajectories.
>
> > **W3:** The authors claim that their application of Kronecker product can facilitate action-state alignment, but there lacks an ablation study to support this claim.
>
> **A**: We agree with the need for an ablation study.  We must first point out that we had a typo in the manuscript. In particular, $\otimes$ represents the outer product rather than the Kronecker product since the state embedding $v \in \mathbb{R}^d$ and the action embedding $m \in \mathbb{R}^d$ are vectors. In other words, for vectors, $\text{OP}(v,m) = \text{KP}(v,m).\text{reshape}(d,d) \in \mathbb{R}^{d \times d}$, where $\text{OP}$ is the outer product operator, and $\text{KP}$ is the Kronecker product operator. Following your suggestion, in the updated appendix, we have added an ablation study to support the advantages of the outer product for action-state alignment compared to the concatenation method as follows.
>
> | Env        | *H*  | Outer Product (DMWM) | Concatenation      |
> |------------|------|-----------------------|---------------------|
> | DMC Tasks  | 10   | 0.729 ± 0.014         | 0.713 ± 0.032       |
> |            | 30   | 0.719 ± 0.042         | 0.697 ± 0.075       |
> |            | 50   | 0.698 ± 0.084         | 0.672 ± 0.128       |
> |            | 100  | 0.687 ± 0.140         | 0.654 ± 0.195       |
> | ManiSkill2 | 10   | 0.721 ± 0.022         | 0.702 ± 0.052       |
> |            | 30   | 0.710 ± 0.049         | 0.681 ± 0.091       |
> |            | 50   | 0.689 ± 0.112         | 0.661 ± 0.183       |
> |            | 100  | 0.669 ± 0.149         | 0.627 ± 0.238       |
> | MyoSuite   | 10   | 0.718 ± 0.031         | 0.698 ± 0.072       |
> |            | 30   | 0.705 ± 0.062         | 0.676 ± 0.125       |
> |            | 50   | 0.682 ± 0.121         | 0.649 ± 0.194       |
> |            | 100  | 0.658 ± 0.163         | 0.617 ± 0.285       |
>
> The simulation results demonstrate that the outer product of DMWM can improve logical consistency compared to the concatenation method across all tasks. In particular, the outer product-based method achieves 3.9% improvements on DMC Tasks, 4.2% on ManiSkill2, and 5.1% on MyoSuite in terms of logical consistency. This is because, compared to the concatenation method that overlooks structural dependencies, the outer product can capture second-order interactions between latent state and action embeddings, which enables cross-space logic modeling.
>
> > **Q:** How did you measure the logical consistency for evaluation? Did you use the trained system II?
>
> **A**: Yes, we use the trained System 2 components to evaluate and verify the logical consistency across different approaches during the test stage. In particular, for a fair comparison, all the LINN-S2  components are trained with the same network architecture, parameter scale, and training steps. For the baselines, including Dreamer, Hieros, and HRSSM, the LINN-S2 components only capture and verify the logic of state transitions and will not be used to impose the logical constraints on the System 1 components during training.

---

> > ### Comment · Reviewer_YoYe · 2025-08-06
> >
> > thank you for your responses and new results. I'm more convinced the paper should be accepted.

---

### Decision · Program_Chairs · 2025-09-17

**Decision:**

Accept (spotlight)

**Comment:**

The paper proposes DMWM, a dual-system world model: System-1 is an RSSM-based predictor; System-2 (LINN-S2) performs neural logical reasoning to enforce logical consistency over imagined trajectories. The two systems interact via inter-system feedback. Experiments on DMControl, ManiSkill2, and MyoSuite evaluate both task return and a logical-consistency metric.

During rebuttal the authors:
(i) clarified the learning-based nature of System-2 (rules are learned from environment traces),
(ii) added ablations motivating the outer/Kronecker product vs. concatenation for state–action alignment,
(iii) explained how logical consistency is measured,
(iv) provided analyses of when the method helps or struggles,
(v) added multi-task and few-shot generalization results, and
(vi) compared exploration efficiency to LS-Imagine, plus expanded related work.

All four reviewers recommend accept after discussion; two explicitly express increased convincing (YoYe, Y1Jx).

Core criticisms (proposed by almost all reviewers), hand-crafted rules and limited logic expressivity, were substantially mitigated by clarifying the learning-based System 2 and by outlining/implementing extensions (probabilistic/first-order).

Methodological clarifications and new experiments (outer/Kronecker ablation; LS-Imagine comparison; multi-task & few-shot generalization; evaluation protocol for logical consistency) directly address the specific asks from reviewers.

Some limitations remain (complexity in highly stochastic settings; broader open-world evaluations), as noted by Jdzu and Y1Jx, but the empirical evidence and clarified scope justify publication.

Overall, this paper is well-organized and high-quality enough for publication.

### Suggestions for the camera-ready
- Retain the explicit “when it works / when it struggles” guidance and training/runtime details in the main text; keep the limitations section prominent.
- Keep the outer/Kronecker vs. concatenation ablation and the logical-consistency evaluation protocol clearly presented.
- Summarize the LS-Imagine comparison and the motivation for applying logic on joint embeddings in the main paper (not only appendix).
- Include the added references and the discussion of probabilistic logic and uncertainty-gated System 2 to make the generalization story concrete.
- Other tips proposed by reviewers.